# Auto-accelerated dehydrogenation of alkane assisted by in-situ formed olefins over boron nitride under aerobic conditions

Zhankai Liu [1], Ziyi Liu[1], Jie Fan[1], Wen-Duo Lu[1], Fan Wu[1], Bin Gao[1], Jian Sheng[1], Bin Qiu[1], Dongqi Wang[1] & An-Hui Lu [1] ✉

Oxidative dehydrogenation (ODH) of alkane over boron nitride (BN) catalyst exhibits high olefin selectivity as well as a small ecological carbon footprint. Here we report an unusual phenomenon that the in-situ formed olefins under reactions are in turn actively accelerating parent alkane conversion over BN by interacting with hydroperoxyl and alkoxyl radicals and generating reactive species which promote oxidation of alkane and olefin formation, through feeding a mixture of alkane and olefin and DFT calculations. The isotope tracer studies reveal the cleavage of C-C bond in propylene when co-existing with propane, directly evidencing the deep-oxidation of olefins occur in the ODH reaction over BN. Furthermore, enhancing the activation of ethane by the in-situ formed olefins from propane is successfully realized at lower temperature by co-feeding alkane mixture strategy. This work unveils the realistic ODH reaction pathway over BN and provides an insight into efficiently producing olefins.

Light olefins, such as propylene and ethylene, are essential raw chemicals for producing various value-added chemical building blocks (e.g. polyethylene, polypropylene, acetone, and acetaldehyde)[1,2]. Currently, the natural gas revolution leads to a switch in the industrial feedstocks, and on-purpose routes for producing olefins from alkanes have emerged, including direct dehydrogenation (DH) reaction[3,4] and oxidative dehydrogenation (ODH) reaction[5,6]. The recent discovery demonstrated that hexagonal boron nitride (h-BN) and boron-containing catalysts with active oxygenated boron species could selectively catalyze the ODH reaction with only negligible CO$_2$ formation, thus opening new avenues toward the selective cleavage of the C−H bond of alkanes[7–16]. Prior studies have reported that the exceptional product distribution on h-BN catalyst is ascribed to the combination of surface-mediated formation of radicals and subsequent gaseous reactions[17–20], which contrasts with the Mars-van Krevelen mechanism on traditional vanadium-based catalysts. However, although some experimental indications of gas-phase reactions were observed[21,22], the complex radical reaction networks hindered the insights into the ODH over h-BN. The understanding of which species

could ensure high alkane conversion and olefin selectivity would pave the way for the construction of a more efficient catalytic reaction system. For example, some highly reactive reactants, which could produce free radicals favorable for the reactions, were introduced to activate alkane at possibly lower temperatures.

Herein, we demonstrated that, during the ODH reaction catalyzed by h-BN, the in-situ formed olefins, which were generally considered chemically inert in boron-based catalytic systems[17,23,24], played a crucial role in promoting parent alkane conversion and cracking of C−C bonds. The experiments of co-feeding mixture of alkane and olefin together with DFT calculations revealed the synergistic reaction routes of the two kinds of hydrocarbons. This unusual observation motivated us to assume that olefins generated from propane could trigger the C−H bond activation of ethane at a lower temperature. Therefore, we conducted a co-feeding strategy of ethane and propane over h-BN catalyst for the ODH reaction. A considerable synergistic conversion of ethane and propane was indeed observed. Kinetic studies evidenced that the synergy was most likely ascribed to the in-situ formed olefins. This co-feeding method of gases mixture will also leave out the

[1]State Key Laboratory of Fine Chemicals, Liaoning Key Laboratory for Catalytic Conversion of Carbon Resources, School of Chemical Engineering, Dalian University of Technology, Dalian 116024 Liaoning, China. ✉e-mail: anhuilu@dlut.edu.cn

procedure of pre-separation before dehydrogenation reaction, significantly reducing the carbon footprint.

## Results

### Accelerating effect of olefins on dehydrogenation of alkanes

A tandem reaction system with two fixed bed reactors (R1 and R2) was designed to investigate the role of the in-situ formed products. As shown in Fig. 1a, in the tandem reaction system, the R1 exhaust, which contained residual reactants ($C_3H_8$ and $O_2$) and the nascent ODH products ($C_3H_6$, $C_2H_4$, CO, and $CO_2$), was fed in the R2 reactor. At 500 °C, the conversions of propane were 9.1% and 12.8% in R1 and R2, respectively, directly evidencing that the propane-derived products could facilitate the propane conversion. Before entering the R2, the exhaust of R1 was passed through a cold trap to remove the $H_2O$. The products formed in the ODH reaction were co-fed with alkane to investigate if the species could facilitate the activation of gaseous alkane. When two gases were co-fed, "A" was designated to represent the main reactant ($CH_4$, $C_2H_6$, or $C_3H_8$) in "A-B" mode with a flow rate of 8 mL min$^{-1}$, and "B" was the added gas ($C_2H_4$, $C_3H_6$, CO, or $CO_2$) with a flow rate of 2.5 mL min$^{-1}$. Figure 1b revealed that no obvious promotion of propane conversion was detected when inletting CO or $CO_2$. Nevertheless, the addition of propylene tremendously enhanced propane conversion from 9.4% to 33.1% at 490 °C. Meanwhile, propylene showed a negative conversion, indicating that the formation rate exceeded the consumption rate of feed-in propylene. When only feeding propylene at the same temperature, almost no conversion of propylene was found. Similar enhanced ethane conversion was also observed in "$C_2H_6$-$C_3H_6$" mode (from 0.9% to 8.0% of ethane conversion at 520 °C). One unanticipated finding was that propylene showed a remarkable conversion of 21.5% (Supplementary Fig. 1), indicating that the activation of propylene did happen under ODH conditions in the presence of ethane. Moreover, "$C_2H_6$-$C_2H_4$", "$C_3H_8$-$C_2H_4$", and "$CH_4$-$C_2H_4$" feeding modes also showed the synergetic activation between alkane and olefin and the detailed data were shown in Supplementary Table 1. For comparison, the R2 inlet gas in the tandem reaction systems with and without added propylene for oxidative dehydrogenation were simulated (Supplementary Fig. 2). When propylene was absent, the propane conversion was 7.2%. The added propylene enhanced propane conversion to 11.7%, which was close to the conversion in R2 measured in the tandem experiment. These outcomes demonstrated that the in-situ formed olefins were the most likely products to enhance the oxidation of alkane. In addition, the promotion effect of olefin with high carbon number was more sensitive and remarkable than that of olefin with low carbon number. We also conducted a "Single-$C_3H_8$ (10.5 sccm)" experiment where the molar ratio of hydrocarbon to oxygen (HC:$O_2$) was the same as that in the "$C_3H_8$-$C_3H_6$" mode. As shown in Supplementary Fig. 3, the $C_3H_8$ conversion was lower than that in "$C_3H_8$-$C_3H_6$" mode and slightly higher than that in "Single-$C_3H_8$ (8 sccm)" mode, indicating that the increased conversion of $C_3H_8$ in "$C_3H_8$-$C_3H_6$" mode was due to the addition of olefin rather than the change of HC:$O_2$ ratio. At 500 °C, propane conversion was 11.6% in "Single-$C_3H_8$ (8 sccm)" mode, lower than that in tandem reaction reactor (20.7%) with the same mass catalyst. The different activity in the two loading modes was caused by the special reaction mechanism of BN which combined with surface and gas-phase reaction. The bed volume and post-catalytic volume were different in single-bed reactor and tandem reactor, resulting in different performances[25,26].

We conducted a series of characterizations to investigate the state of catalysts before and after ODH reaction under different atmospheres. Four catalysts were characterized: fresh BN, activated BN, BN after reaction in "Single-$C_3H_8$" mode for 3 h (Single-$C_3H_8$-BN), and BN after reaction in "$C_3H_8$-$C_3H_6$" mode for 3 h ($C_3H_8$-$C_3H_6$-BN). The specific treatment processes for catalysts were shown in Supplementary Table 2. The FT-IR spectra showed the band at around 3400 cm$^{-1}$ assigned to O−H vibration appeared on the activated BN (Supplementary Fig. 4)[7]. The X-ray diffraction (XRD) pattern of activated BN exhibited a new diffraction peak at $2\theta = 14.6$ °, corresponding to the generation of $BO_x$. (Supplementary Fig. 5)[13]. In the $NH_3$-TPD experiment, activated BN showed a new $NH_3$ desorption peak, which also proved the existence of $BO_x$. (Supplementary Fig. 6). The SEM showed that the boron nitride sheets changed from uniform dispersion in fresh BN to aggregation in activated BN (Supplementary Fig. 7). These characterizations indicated the formation of active species under the ODH conditions. When the activated BN were treated for another 3 h under the two atmospheres ("Single-$C_3H_8$", "$C_3H_8$-$C_3H_6$"), the surface functional groups and morphology were almost unchanged and no new species were regenerated. The above results demonstrated that the activated BN was stable and not affected by the composition of the reaction gas. The content of $BO_x$ species in these materials were similar, indicating that the enhanced activity in co-feeding mode was not caused by the increase of active sites.

Temperature-programmed experiment with "$C_2H_6$-$C_3H_6$" feeding mode was conducted to further illustrate the synergistic effect between alkane and olefin. As shown in Fig. 2a, $C_3H_6$ started to be consumed at 454 °C, being lower than that in "Single-$C_3H_6$" where the activation temperature of $C_3H_6$ was 492 °C (Supplementary Fig. 8). $C_2H_4$ and CO were simultaneously detected at around 454 °C, demonstrating that these two products were generated from $C_3H_6$ rather than $C_2H_6$ at the lower temperature. The signal strength of $C_2H_6$ began to decrease when the temperature continued to rise to 478 °C, which was not observed in "Single-$C_2H_6$" before the temperature reached 499 °C (Supplementary Fig. 9). These results indicated that the activation temperature of reactants could be significantly decreased by co-feeding olefin with alkane. Furthermore, the signal intensity of $C_2H_4$ reached the maximum at 555 °C and then sharply decreased along with an increase in the consumption rate of $C_2H_6$ in "Single-$C_2H_6$", demonstrating a co-activation between ethane and ethylene.

To probe the relationship between alkane and olefin in co-feeding atmosphere, catalytic tests with various $p_{C2H6}/p_{C3H6}$ (the ratio of partial pressure of ethane to partial pressure of propylene) were operated. As shown in Supplementary Fig. 10a, ethane and propylene reacted in different stoichiometric ratios under different $p_{C2H6}/p_{C3H6}$. With the

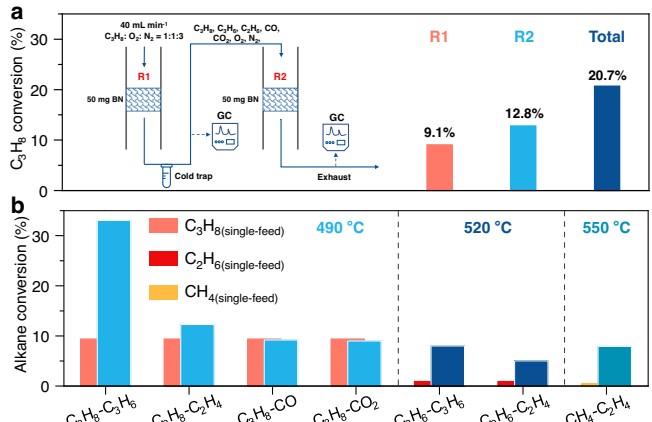

**Fig. 1 | Evidence for olefins accelerating alkane conversion. a** The conversion of propane in the tandem reaction system at 500 °C. $F_{total} = 40$ mL min$^{-1}$, $C_2H_6$:$C_3H_8$:$O_2$:$N_2$ = 8:2.5:8:21.5. **b** Alkane conversion in different "Single-alkane" and "alkane-olefin" feeding modes. The total flow velocity of 40 mL min$^{-1}$ with $O_2$ flow velocity of 8 mL min$^{-1}$ was employed in all the catalytic tests.

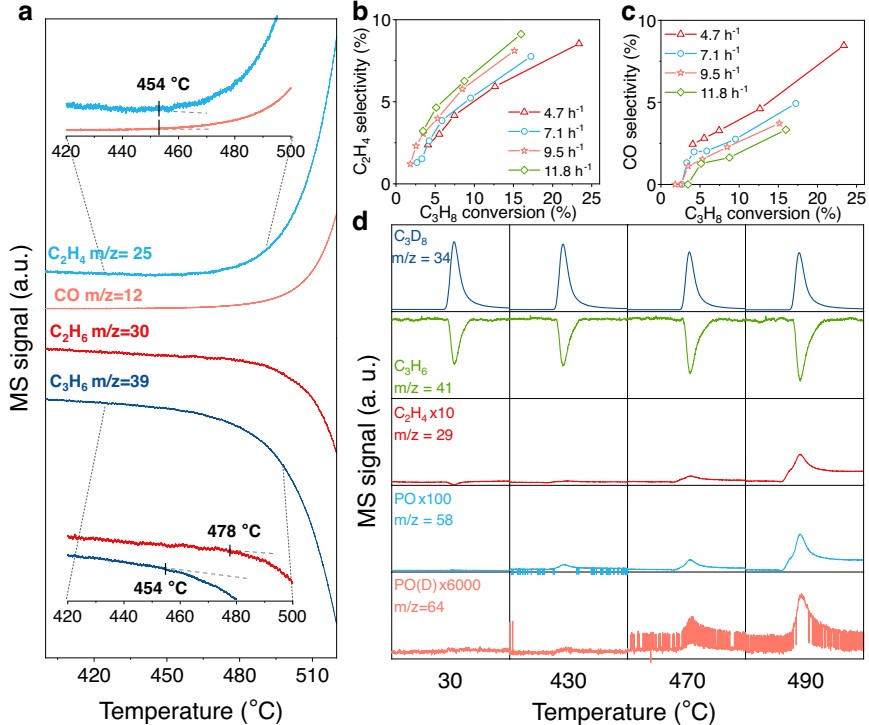

**Fig. 2 | Key species in the ODH reaction system. a** "C$_2$H$_6$–C$_3$H$_6$" temperature-programmed reaction. Signals of C$_2$H$_4$ ($m/z = 25$), CO ($m/z = 12$), C$_2$H$_6$ ($m/z = 30$), and C$_3$H$_6$ ($m/z = 39$) were measured via mass spectrometer. $F_{total} = 40$ mL min$^{-1}$, C$_2$H$_6$:C$_3$H$_6$:O$_2$:N$_2$ = 8:2.5:8:21.5. (**b, c**) C$_2$H$_4$ and CO selectivity as function of propane conversion at various WHSV. C$_3$H$_8$: O$_2$: N$_2$ = 8: 8: 24; reaction temperature, 410–510 °C. **d** Gas profile in transient pulses of C$_3$D$_8$ over BN. Signals of C$_3$D$_8$ ($m/z = 34$), C$_3$H$_6$ ($m/z = 41$), C$_2$H$_4$ ($m/z = 29$), PO ($m/z = 58$), and PO(D) ($m/z = 64$) were measured via mass spectrometer. $F_{total} = 40$ mL min$^{-1}$, C$_3$H$_6$:O$_2$:N$_2$ = 2.5:8:29.5; m$_{cat}$ = 100 mg; pulse value: 1 mL each time.

increase of $p_{C2H6}/p_{C3H6}$, $r_{C2H6}/r_{C3H6}$ (the ratio of reaction rate of ethane to that of propylene) became higher, showing a more significant promotion effect of propylene on the activation of ethane. This is highly possible because, in a high ethane concentration atmosphere, it is easier for ethane molecules to access active intermediate species generated by propylene. A similar pattern was also observed in the experiment when changing $p_{CH4}/p_{C2H4}$ (Supplementary Fig. 10b). However, $r_{CH4}$ was suppressed with the increase of $r_{C2H4}$ when $p_{CH4}/p_{C2H4}$ was 2.5/8 and even showed a negative conversion, indicating that the formation rate of methane from ethylene exceeded the consumption rate of feed-in methane.

In addition, the ODH reactions of propane at varying weight hourly space velocity (WHSV) in the temperature range of 410–510 °C were operated to evaluate the product distribution. The propane conversion was higher at lower WHSV (Supplementary Fig. 11a), indicating more consumption of propane at a longer contact time. As the WHSV decreased, a decrease in the selectivity of ethylene and an increase in the selectivity of CO at the same propane conversion was observed (Fig. 2b, c). It may be ascribed to that more part of the propane conversion was contributed by the interaction between ethylene and propane at a lower WHSV, leading to more ethylene conversion to CO. However, the propylene selectivity was insensitive to the change in WHSV (Supplementary Fig. 11b), which may be because propane could produce additional propylene to compensate for the reduced selectivity when interacting with olefins, resulting in almost constant selectivity of propylene. The "C$_3$H$_8$–C$_3$H$_6$" experiments with different WHSVs were also conducted to evaluate the influence of WHSV on the product distribution (Supplementary Fig. 12). In these experiments, since part of the co-fed propylene was consumed to promote the conversion of propane, the calculated propylene selectivity offers a lower limit for the propylene production. The outcomes showed that the selectivity of ethylene and CO

increased and propylene selectivity decreased when decreasing WHSV at a constant propane conversion.

## Mechanism of the interaction between alkane and olefin

Previous studies reported that the oxidation rate of hydrocarbon and the selectivity to propylene oxide (PO) were enhanced by mixing propane and propylene without catalysts[27,28]. In addition, the formation of PO in the ODH reaction of propane over BN was also detected in recent studies[29,30] and current work. Given this, here we put forward a hypothesis that PO was an intermediate product formed by the interaction of propane and propylene in the ODH reaction of propane catalyzed by BN. To corroborate the hypothesis, the pulsed propylene (1 mL each time) experiment was performed in the propane ODH reaction atmosphere at different temperatures, and the signal of PO was measured by using mass spectrometer. Supplementary Fig. 13 shows that the signal intensity of pulsed propylene gradually decreased with elevated temperature and the signal of PO was clearly observed when the temperature reached 430 °C, suggesting that the introduction of the additional propylene promoted the formation of PO.

To clarify whether PO was formed from propane or propylene, pulses of a small amount of deuterated propane (C$_3$D$_8$) were added into a constant flow of C$_3$H$_6$, O$_2$, and N$_2$ at different temperatures. As shown in Fig. 2d, the consumption of C$_3$H$_6$ and C$_3$D$_8$ increased with the increased temperature, indicating that the stronger synergy between C$_3$D$_8$ and C$_3$H$_6$ occurred at a higher temperature. And C$_2$H$_4$ was gradually generated along with the increased consumption of C$_3$D$_8$ as the temperature elevated from 30 °C to 490 °C. This result directly evidenced the cleavage of the C–C bond in propylene during the co-reaction of the two kinds of hydrocarbons. It also confirmed the previous viewpoint that the decrease in propylene selectivity with increasing propane conversion was indicative of the facile

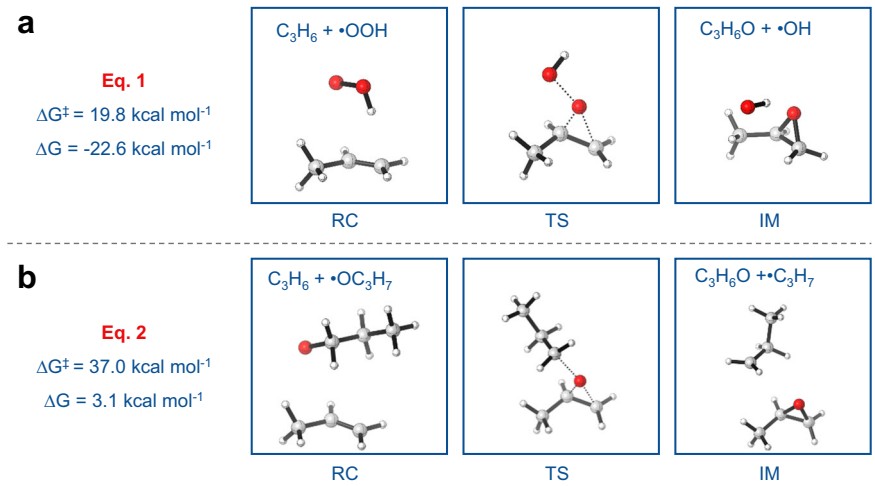

**Fig. 3 | Possible routes for activating propylene.** Key stationary points and relative free energies ($\Delta G^{\ddagger}$ and $\Delta G$, in kcal mol$^{-1}$) in (**a, b**) the two processes responsible for PO generation.

overoxidation of propene[7]. In addition, the signals of PO and deuterated PO [PO(D)] were also detected when the temperature was above 430 °C, indicating that PO could be produced from both propylene and propane. A small shoulder at an earlier reaction time that precedes the addition of $C_3D_8$ and consumption of $C_3H_6$ may be a signal fluctuation by inserting the syringe into the channel prior to pulse. Furthermore, as shown in Supplementary Fig. 14, the signal of $C_3H_6$ consumption and PO generation appeared simultaneously, indicating PO may be the primary product in the oxidation reaction of $C_3H_6$. The PO(D) formation signal delayed the $C_3H_6$ consumption signal by 8.4 s, implicating that PO(D) may be the secondary product formed from $C_3D_8$-derived $C_3D_6$. The same experiment was performed just by changing propylene to ethylene and the outcome showed that the intensity of $C_3D_8$ and PO(D) exhibited a trend similar to the above measurement (Supplementary Fig. 15). Likewise, the delayed signal of PO(D) was also observed (Supplementary Fig. 16). In contrast, PO was not formed with the consumption of $C_2H_4$, except for the observed fluctuations caused by the pulses, demonstrating that PO could generate only from $C_3$ species.

Considering highly active PO was unstable at high temperature in the atmosphere containing gaseous oxygen, we deduced that propylene-derived PO would be further transformed into other species. Thus, PO oxidation tests were carried out at high temperatures to investigate the product distribution. At 450 °C and 510 °C, the main products were CO and ethylene, together with a small amount of $CO_2$ (Supplementary Fig. 17), which were similar to the products from the oxidation of propylene. These results support the hypothesis that PO was an intermediate formed from propylene in the ODH reaction of propane. In addition, due to the undetectable adsorption of propane and propylene on BN (Supplementary Fig. 18), we deduced that the synergistic effect of the two hydrocarbons occurred in the gas-phase. Moreover, "$C_3H_8$–$C_3H_6$" blank test without catalyst was conducted to investigate the effect of gas phase. As shown in Supplementary Fig. 19, the empty reactor also showed the synergy between propane and propylene, which was however much weaker than that with BN catalyst. Therefore, the synergy was likely to occur in the gas phase and was enhanced by BN. Given the $BO_x$ species were considered to be able to produce highly reactive radicals and release them to gas phase[18,25], the $BO_x$ sites formed on the BN surface may play a role in increasing the concentration of active species with the assistance of oxygen, e.g. propyl peroxide ($C_3H_7OOH$), which may be easily formed by rebound of peroxyl and propyl radicals[19] and then released to the gas phase to react with propylene.

Density functional theory (DFT) calculations were carried out to understand the role of co-fed propylene on the ODH of propane. In our previous study[19], we showed that $C_3H_7OOH$ could be produced in the ODH of propane catalyzed by boron-based catalysts. Here we first studied the potential role of propylene on the decomposition of $C_3H_7OOH$. The decomposition of $C_3H_7OOH$ generated propoxy (•$OC_3H_7$) and hydroxyl radical (•OH) with an exergonicity of 36.0 kcal mol$^{-1}$. In order to evaluate the energy required to break the peroxo bond, the decomposition in both singlet and triplet states were calculated, and a crossing point was located at the peroxo bond ($C_3H_7O$−OH) distance of 2.0 Å with an energy of 46.6 kcal mol$^{-1}$ relative to the starting state of propyl peroxide (Supplementary Fig. 20). This value showed that it was not easy to break the alkyl peroxo bond under thermodynamic condition. A molecule of propylene was then considered to participate in the thermal decomposition of the peroxo bond (Fig. 3a), and the barrier was found to decrease to 19.8 kcal mol$^{-1}$, showing the significant acceleration in the presence of propylene. This reaction produced a PO molecule and •OH, which was more reactive than hydroperoxyl radical (•OOH).

In the oxidative environment of ODH, there are various oxidants. Here we consider two relatively stable oxidants that are also relevant to the deep oxidation of alkanes, •$OC_3H_7$ and •OOH, and study how the co-feeding of propylene may help their clearance (Fig. 3). In our previous work[19], we showed the •$OC_3H_7$ may be eliminated through reaction with the surface >B-OH site. In the presence of propylene, it is also possible for •$OC_3H_7$ to react with propylene, which is moderately endergonic by 3.1 kcal mol$^{-1}$ with a free energy barrier of 37.0 kcal mol$^{-1}$ according to our calculations (Fig. 3b), showing that this process is thermodynamically more favorable but kinetically less favorable than its elimination on the surface >B-OH site. This indicates that under ODH condition, the reaction of •$OC_3H_7$ with propylene may offer another way to eliminate the gaseous •$OC_3H_7$ species in addition to the surface >B-OH pathway. The reaction of propylene with •$OC_3H_7$ produces PO, which has been detected in experimental studies, and complicates the mechanisms by introducing the elementary reactions of the degradation of PO. In earlier studies, gaseous PO was oxidized to acrolein and CO at the temperature above 200 °C[31]. According to our experimental measurement, co-feeding of propylene results in a moderate increase of CO formation (Supplementary Fig. 21). Concerning the complexity of the gaseous channels with the presence of various reactive gaseous species, it is hard to quantitatively estimate the contribution of the PO degradation to the formation of CO.

According to our calculations, the reaction of propylene with •OOH is another process responsible for the formation of PO. The

barrier to this step is 19.8 kcal mol$^{-1}$ with an exergonicity of 22.6 kcal mol$^{-1}$, indicating that the •OOH may be efficiently eliminated once colliding with propylene under the ODH condition. This benefits the ODH reaction since it produces an oxidant, •OH, which is more reactive than •OOH. As a precursor to the formation of •OOH and •OC$_3$H$_7$ radical, propane affects the oxidation of propylene to PO.

Based on the analysis of data from experimental measurement and calculations, we proposed the reaction routes to show how the coexistence of propylene may influence the ODH reaction of alkane (Fig. 4). Because the difference between primary and secondary C–H bond was relatively small, the more widely exposed primary C–H bond breaking was adopted (Supplementary Fig. 22). The hydroperoxyl and alkoxyl radicals may react with propylene to generate more reactive hydroxyl radical and

alkyl radical. The barriers to the formation of n-alkoxyl and iso-alkoxyl radicals via the reaction of •OOH radical with the primary and secondary propyl radicals, respectively, were shown in Supplementary Fig. 23, which were much lower than that to the activation of alkane. The hydroxyl radical favors the oxidation of alkane, and the evolution of alkoxyl to alkyl radical guides the ODH reaction back to the track of olefin formation rather than the deep oxidation. After reaction, propylene is converted to PO, which was then further oxidized to ethylene and CO. The generated ethylene could also react with hydroperoxyl and alkoxyl radicals. Therefore, we deduced that some ethylene and CO are generated from propylene via interacting with propane-derived species in the ODH reaction of propane.

### Performances of BN for co-feeding alkanes mixture

Based on such unexpected discovery over BN catalyst, the co-feed experiment of using propane and ethane was conducted to realize the activation of ethane by in-situ formed olefins from propane at lower temperatures. The recent work by Xu et al. also showed that the introduction of propane was able to enhance the ethane conversion and proposed propane as a radical generator[29]. Figure 5a shows that co-feeding of a small amount of propane (2.5 mL min$^{-1}$) with ethane (8 mL min$^{-1}$) exhibited promotion of the conversion of both at all tested reaction temperatures. When

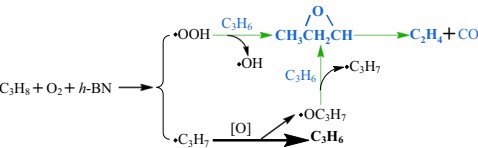

**Fig. 4 | Mechanistic insights for the synergy between propane and propylene.** Reaction routes associated with the gaseous interactions between propane and propylene in the ODH reaction.

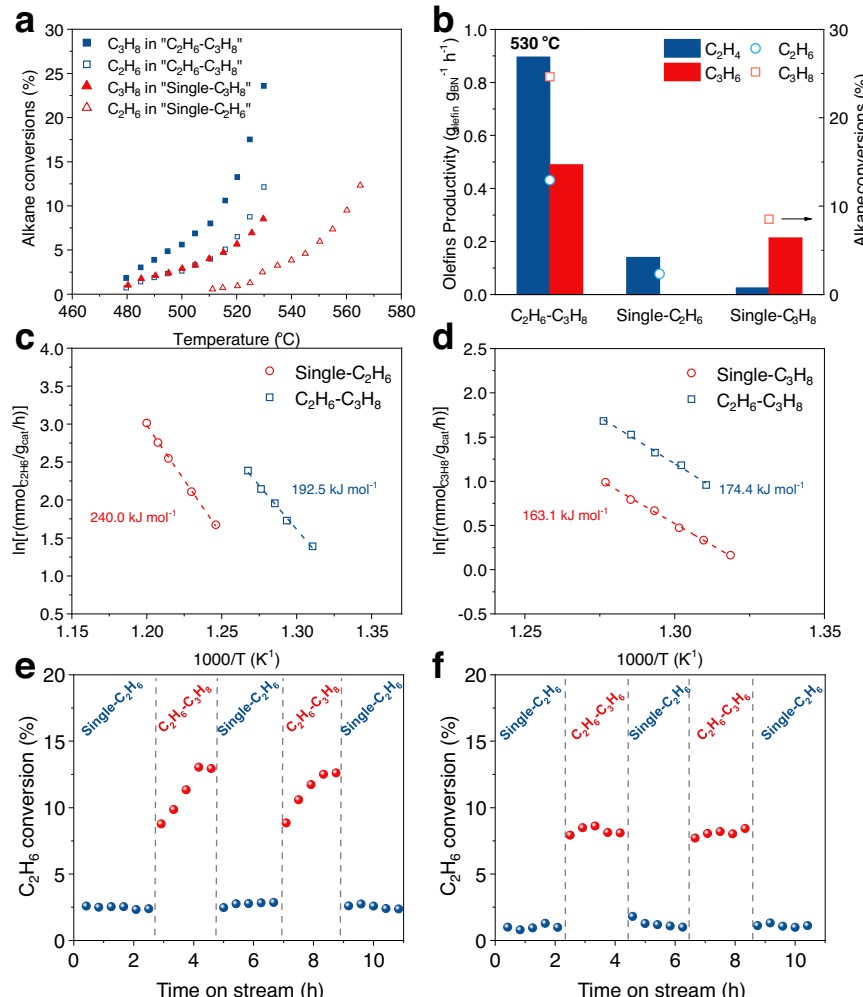

**Fig. 5 | Synergistic conversion between ethane and propane. a** Dependence of alkane conversions on temperature over BN in different feeding modes. **b** Alkane conversions and olefins productivity in different feeding modes over BN at 530 °C. Arrhenius plots for the reaction rate (apparent activation energy, $E_a$) of **c** ethane and **d** propane over BN. Ethane conversion as a function of time-on-stream during the cycles of **e** "Single-C$_2$H$_6$" or "C$_2$H$_6$–C$_3$H$_8$" and **f** "Single-C$_2$H$_6$" or "C$_2$H$_6$–C$_3$H$_6$" over BN. $F_{total}$ = 40 mL min$^{-1}$, "Single-C$_2$H$_6$", C$_2$H$_6$:O$_2$:N$_2$ = 8:8:24; "Single-C$_3$H$_8$", C$_3$H$_8$:O$_2$:N$_2$ = 2.5:8:29.5; "C$_2$H$_6$–C$_3$H$_8$/C$_3$H$_6$", C$_2$H$_6$:C$_3$H$_8$/C$_3$H$_6$:O$_2$:N$_2$ = 8:2.5:8:21.5.

the same ethane conversion was achieved, the temperature required for the "$C_2H_6$–$C_3H_8$" mode was approximately 35 °C lower than that of the "Single-$C_2H_6$" mode. At 530 °C, the conversion of ethane was 13.0% with ethylene productivity of 0.90 $g_{C2H4}$ $g_{cat}^{-1}$ $h^{-1}$ under co-feeding atmosphere, a nearly 5-fold increase over that under ethane single-feed atmosphere (Fig. 5b). However, the capacity of propane for promoting ethane conversion was lower than that of propylene (26.5% conversion of ethane at 530 °C), indicating that propane may facilitate ethane activation through the formed olefins under reaction conditions. Meanwhile, the conversion of propane and the productivity of propylene increased from 8.5% and 0.21 $g_{C3H6}$ $g_{cat}^{-1}$ $h^{-1}$ to 24.7% and 0.50 $g_{C3H6}$ $g_{cat}^{-1}$ $h^{-1}$, respectively. The alkane conversions and product selectivity in the "$C_2H_6$-$C_3H_8$" mode were shown in Supplementary Fig. 24. The main products were ethylene and propylene, and with the increase in temperature, the selectivity of propylene decreased and ethylene increased. The product selectivity and space-time yield (STY) at similar conversions of alkane in the two feeding modes were compared (mode 1: "$C_2H_6$-$C_3H_8$"; mode 2: "Single-$C_2H_6$" + "Single-$C_3H_8$"). As shown in Supplementary Table 3, the selectivity of olefins and STY in the co-feeding mode was essentially same as that in the single-feeding mode. These data showed that the co-feeding mode offered high conversion at lower reaction temperature without significantly changing the selectivity of olefins. The co-feeding tests were also applied to evaluate the catalytic activity of the reference catalysts, i.e. $Al_2O_3$ supported $VO_x$ (V–$Al_2O_3$) and Li-doped MgO (L–MgO), in comparison with that of the BN catalyst. The XRD patterns of the three catalysts were shown in Supplementary Fig. 25. These efficient ODH catalysts have been well studied in literature and considered to follow different mechanisms[32,33]. For the reference catalysts, the co-feeding of propane only promoted the conversion of ethane less than 2 times and ethane did not enhance the activation of propane (Supplementary Fig. 26). The unique synergistic activation of propane and ethane over BN was attributed to the mechanism with the co-presence and co-operation of surface-mediated radical formation processes and gas-phase channels, which was different from the Mars-van Krevelen mechanism of redox metal oxides and adsorbed oxygen mechanism of rare-earth oxide. Possibly due to the dominance of homogeneous reaction initiated by radical at the high temperature, the synergistic effect of the two hydrocarbons was observed on Li-MgO when the temperature was above 570 °C (Supplementary Fig. 27). In addition to the exceptional high conversion of alkanes, another merit for BN was the high olefins selectivity. It afforded 97.6% selectivity to olefins at 15.5% alkanes conversion, significantly higher than the reference catalysts (Supplementary Fig. 28). We conducted a "Single-$C_2H_6$ (10.5 sccm)" experiment where the HC:$O_2$ ratio was the same as that in the "$C_2H_6$–$C_3H_8$" mode (Supplementary Fig. 29). In this scenario, the $C_2H_6$ conversion was lower than that in "$C_2H_6$–$C_3H_8$" mode and slightly higher than that in "Single-$C_2H_6$ 8 sccm" mode, indicating that the increased reactivity of $C_2H_6$ in "$C_2H_6$–$C_3H_8$" mode was due to the addition of another alkane rather than changing the HC:$O_2$ ratio. Furthermore, we conducted "$C_2H_6$–$C_3H_8$" blank test without any catalyst (Supplementary Fig. 30). The conversion of ethane was 7.1% at 570 °C, only a twofold increase over that in the "Single-$C_2H_6$" mode. These results demonstrated the role of BN to decrease reaction temperature and boost the synergistic effect.

The selectivity of ethane-derived ethylene as a function of ethane conversion under the two feeding modes was shown in Supplementary Fig. 31. Under the co-feeding atmosphere, the selectivity of ethylene was obtained by fitting, assuming that the correlation between the product distribution of ODH of propane and the conversion of propane was unchanged under the two feeding modes. Over BN, the presence of propane promoted the generation of CO, reducing the selectivity of ethylene. This result confirmed that in different feeding modes, alkanes conversion over BN catalyst followed different reaction mechanisms. According to the above study, CO was most likely to be produced from olefins by reacting with alkanes. In addition, the co-fed propane significantly altered the apparent activation energy ($E_a$) of ethane for BN (Fig. 5c). However, probably because the ethane-derived ethylene which activated propane could be also generated from ODH of propane, or the capacity of ethylene for promoting activation was lower than that of propylene, similar $E_a$ of propane was shown in the two feeding modes (Fig. 5d).

To assess the difference between propane and propylene in promoting ethane conversion, the reversibility of ethane conversion was studied over BN catalyst through feeding "$C_2H_6$-$C_3H_8$" and "Single-$C_2H_6$" alternatively over a period of 650 min at 530 °C. Figure 5e shows that the conversion of ethane remained constant at about 2.7% in "single-$C_2H_6$", and rapidly increased to 8.8% within 25 min after switching to "$C_2H_6$-$C_3H_8$". The value reached 13.0% within the next 75 min and remained steady during the co-feeding of "$C_2H_6$-$C_3H_8$". Once switching back to "Single-$C_2H_6$" again, the conversion of ethane went down to 2.5% shortly. Remarkably, the added propane may not lead to irreversible positive or negative effects over BN catalyst because similar ethane conversions were obtained under three cycles of "Single-$C_2H_6$". The cycles of "Single-$C_2H_6$" or "$C_2H_6$-$C_3H_8$" were also operated at 520 °C (Fig. 5f). Different from the relatively slower promotion in the cycles of "Single-$C_2H_6$" and "$C_2H_6$-$C_3H_8$", the conversion of ethane rapidly increased to 8.0% within 25 min and remained steady when the feedstock was switched from "Single-$C_2H_6$" to "$C_2H_6$-$C_3H_6$". Therefore, the slow increase in ethane conversion during the cycles of "Single-$C_2H_6$" and "$C_2H_6$-$C_3H_8$" may be attributed to the lower concentration of olefins during the initiation stage in the reaction tube.

The "$CH_4$-$C_2H_6$" mode was also conducted over BN catalyst (Supplementary Fig. 32) to evaluate whether there is also a synergistic phenomenon with the other alkane combinations. When ethane was absent, less than 1% conversion of methane was observed even above 610 °C. Ethane with 2.5 mL $min^{-1}$ tremendously enhanced the conversion of methane (8 mL $min^{-1}$) to 20.5% at 600 °C, whereas the conversion of ethane was not affected by methane. It was worth noting that ethylene also promoted methane conversion more than ethane at the same temperature, indicating that olefins exhibited stronger activation capacity than alkanes with the same carbon number. According to previous studies[34,35], olefins were the main products in the ODH reactions of ethane or propane over boron-based catalysts, while only a trace number of olefins were formed in the methane oxidation reaction. These outcomes emphasized the pivotal contribution of the additional olefins in promoting the ODH of alkanes, and may explain why methane did not promote the conversion of ethane in "$CH_4$-$C_2H_6$".

In addition to olefins, $H_2O$ has been shown to improve the BN-catalyzed ODH performance, which is another product in the reaction[18]. Because two stages of rapid and slow changes of alkane conversion were observed in the cyclic experiment, two possible roles of $H_2O$ were proposed to generate free radicals and increase the concentration of active surface species. Our previous work on calculation also supported that $H_2O$ could interact with the surface to form more B-OH[19]. Compared with $H_2O$, propylene only showed a rapid change of alkane conversion in the cyclic experiment. In addition, the presence of olefins may enhance the deep-oxidation route while $H_2O$ did not. Therefore, although both species demonstrated the promotional effect on alkane conversion, they differed in enhancing reaction routes and changing the species or concentration of free radicals and surface active sites. These observations suggest that the complex surface-gas-phase

reaction involved many synergies and it is worth further investigation in the future to comprehend this system.

## Discussion

In summary, a series of co-feeding experiments of alkane and olefin revealed that in-situ formed olefins played a vital role in accelerating the conversion of parent alkanes in the ODH reaction over BN catalysts. Combining isotope tracer studies and DFT calculations, we constructed the synergistic reaction routes of alkane and olefin in the ODH catalyzed by BN catalyst. Furthermore, an important conclusion was proposed that some by-products (ethylene and CO) were produced via cracking the C–C bond of propylene-derived PO in the ODH reaction of propane. Finally, an efficient strategy to conduct ODH reaction over BN catalysts by feeding alkane mixture was applied, which afforded a remarkable promotion of alkane conversions with the aid of in-situ formed olefins.

## Methods

### Materials

Hexagonal boron nitride (*h*-BN), P123, hydrophilic silicon dioxide ($SiO_2$), propylene oxide (PO), and aluminum nitrate nonahydrate ($Al(NO_3)_3 \cdot 9H_2O$), were purchased from Aladdin Industrial Inc. Tetraethoxysilane (TEOS), hydrochloric acid (HCl 12 M), Mg powder, boric acid ($H_3BO_3$), lithium carbonate ($Li_2CO_3$), magnesium oxide (MgO), and urea ($CO(NH_2)_2$) were supplied from Sino-Pharm Chemical Reagent Co. Ltd. Ammonium metavanadate ($NH_4VO_3$) were supplied by Tianjin Guangfu Technology Development Co. Ltd. Alkane gases ($CH_4$, $C_2H_6$, $C_3H_8$, $i$-$C_4H_{10}$), olefin gases ($C_2H_4$, $C_3H_6$), $O_2$, Ar, $N_2$, carbon monoxide (CO), and carbon dioxide ($CO_2$) were supplied by Dalian Special Gas Co. Ltd. Deuterated propane ($C_3D_8$) was supplied by Aldrich Chemistry.

### Catalysts synthesis

Treatment of boron nitride (BN). Before use, 2 g of *h*-BN was dispersed in 200 mL of deionized water and stirred at 80 °C for 2 h, then the sample was filtrated, dried at 100 °C for 12 h.

Synthesis of $Al_2O_3$ supported $VO_x$ (V-$Al_2O_3$). $Al_2O_3$ was synthesized via the hydro-thermal method with $Al(NO_3)_3 \cdot 9H_2O$ and $CO(NH_2)_2$ with a molar ratio of 1:5 as raw materials[36]. V-$Al_2O_3$ was prepared via incipient wetness impregnation method with $Al_2O_3$ as support. The impregnation was accomplished by dissolving $NH_4VO_3$ in an aqueous solution of oxalic acid. After impregnation, the sample was dried at 90 °C for 12 h and then calcined at 500 °C for 2 h and V-$Al_2O_3$ was obtained.

Synthesis of Li-doped MgO (Li-MgO). Li-MgO was prepared by the addition of 2.5 g of MgO and 1.5 g of $Li_2CO_3$ to 100 mL of deionized water. The mixture was adequately stirred at 60 °C for 4 h and vacuum-filtered. The acquired precipitate was dried at 60 °C and then calcined at 800 °C for 4 h with a heating rate of 2 °C/min in air to prepare Li-MgO.

### Catalytic testing

The oxidative dehydrogenation (ODH) reaction tests were performed in a continuous flow packed-bed quartz tube (i.d.= 9 mm, 420 mm in length) under atmospheric pressure (0.1 MPa). Catalyst (BN: 100 mg, V−$Al_2O_3$: 40 mg, Li−MgO: 100 mg) with 80–100 mesh was placed in the constant temperature zone of a reactor. The feed gases including $CH_4$ (99.9%), $C_2H_6$ (99.9%), $C_2H_4$ (99.9%), $C_3H_8$ (99.9%), $C_3H_6$ (99.9%), $O_2$ (99.99%), and $N_2$ (99.999%) were controlled separately by mass flow controllers to vary total flows and partial pressures of the reactants. The reaction temperature was controlled by a thermocouple placed at the inner center of the catalyst bed. Before evaluation of the catalytic activity, BN, V−$Al_2O_3$, and Li−MgO were pretreated for 3 h under reaction atmosphere ($F_{total}$ = 40 mL min$^{-1}$, $C_3H_8$:$O_2$:$N_2$ = 8:8:24) at 550 °C, 500 °C, and 600 °C, respectively. Reactants and products were

analyzed by an on-line gas chromatograph (Techcomp, GC 7900). AGDX-102 and 5 A molecular sieve columns, connected to a TCD were used to analyze alkane conversion and products selectivity, which were calculated according to equations as follows. Carbon balances were always within the range of 100 ± 5%.

### Equations

$$ni = \frac{A_i}{\text{RF}_i} \tag{1}$$

Where $n_i$ is the absolute moles of a component "$i$", $A_i$ is the peak area directly obtained by GC, $\text{RF}_i$ is the response factor.

Expansion factor (EF):

$$\text{EF} = \frac{n_{N_2}^{\text{out}}}{n_{N_2}^{\text{in}}} \tag{2}$$

(1) Equations in single-feeding mode.
Conversion of alkane ($X$, %):

$$X = \left(1 - \frac{n_{\text{alkane}}^{\text{out}}}{\text{EF}^* n_{\text{alkane}}^{\text{in}}}\right) \times 100 \tag{3}$$

Selectivity of hydrocarbon product "$i$" on a carbon basis ($S_i$, %):

$$S_i = \frac{100 \times n_i^{\text{out}} \times N_i^{\text{carbon}}}{(\text{EF} \times n_{CnH2n+2}^{\text{in}} - n_{CnH2n+2}^{\text{out}}) \times n + (\text{EF} \times n_{CmH2m+2}^{\text{in}} - n_{CmH2m+2}^{\text{out}}) \times m} \tag{4}$$

Space-time yield ($\text{STY}_i$, mmol $g_{\text{cat}}^{-1}$ h$^{-1}$):

$$\text{STY}_i = \frac{F_{\text{alkane}} \times X \times S_i \times 60}{m_{\text{cat}} \times 22.4 \times (\text{carbon number of olefin})} \tag{5}$$

Where $F_{\text{alkane}}$ is the velocity of alkane, mL min$^{-1}$.

(2) Equations in co-feeding mode.
Mol % of component $i$ in co-feeding mode ($M_i$, %):

$$M_i = \frac{100 \times n_i}{\sum_i n_i} \tag{6}$$

Conversion of alkanes mixture on a carbon basis ($X_{\text{carbon}}$, %):

$$X_{\text{carbon}} = \frac{M_{CnH2n+2} \times X_{CnH2n+2} \times n + M_{CmH2m+2} \times X_{CmH2m+2} \times m}{M_{CnH2n+2} \times n + M_{CmH2m+2} \times m} \times 100 \tag{7}$$

Yield of olefin ($Y_i$, %):

$$Y_i = X_{\text{carbon}} \times S_i \tag{8}$$

Space-time yield ($\text{STY}_i$, mmol $g_{\text{cat}}^{-1}$ h$^{-1}$):

$$\text{STY}_i = \frac{(F_{CnH2n+2} \times n + F_{CmH2m+2} \times m) \times X_{\text{carbon}} \times S_i \times 60}{m_{\text{cat}} \times 22.4 \times (\text{carbon number of olefin})} \tag{9}$$

(3) Conversion of propane in R2 of tandem reactor ($X$, %):

$$X_{C3H8(R2)} = \left(1 - \frac{n_{R2}^{\text{out}}}{\text{EF} \times n_{C3H8(R2)}^{\text{in}}}\right) \times 100 = \left(1 - \frac{n_{R2}^{\text{out}}}{\text{EF} \times n_{C3H8(R1)}^{\text{out 500 °C}}}\right) \times 100 \tag{10}$$

The kinetic data (the reaction orders and apparent activation energies) were measured with alkane conversion below 12%. Using Arrhenius equation to determine apparent activation energies of

alkane conversion ($E_a$):

$$-r_{\text{alkane}} = A \times e^{-E_a/RT} \qquad (11)$$

The $E_a$ was calculated based on the linear correlation between ln $r_{\text{alkane}}$. and $1/T$.

## Characterization

Fourier-transform infrared spectroscopy (FT-IR) spectrum was measured in the transmittance mode by a Nicolet 6700 spectrometer using a mercury cadmium telluride (MCT). The as-prepared samples were prior milled with KBr and compressed into a pellet. Powder X-ray diffraction (XRD) was conducted with a PANalytical X'Pert3 Powder diffractometer to analyze the structure of catalysts. The radiation source was a monochromatic Cu Kα (λ = 0.15406 nm) with an operating condition at 40 kV and 40 mA. The acid sites on catalysts were determined by NH$_3$-temperature-programmed desorption-mass spectrometry (NH$_3$-TPD) on a Micromeritics AutoChem II 2920 apparatus. 100 mg catalyst was loaded into a U-shaped quartz tube and pretreated at 400 °C for 1 h in He, then cooled to 100 °C in the inert atmosphere. At 100 °C, NH$_3$ (1 mL each time) was directly pulsed into the catalyst using He (30 mL min$^{-1}$) as the carrier gas. The weakly adsorbed NH$_3$ were removed by sweeping pure He at 100 °C for 0.5 h. Then the TPD measurement was conducted over the range 100–400 °C at a heating rate of 10 °C/min in a He flow. Signals for NH$_3$ ($m/z$ = 17, 16, 15) were monitored using on-line mass spectrometry (MS). The scanning electron microscopy (SEM) investigation was carried out with a Hitachi FESEM SU8220 instrument. C$_3$H$_8$ or C$_3$H$_6$ temperature-programmed desorption (TPD) was conducted on a Micromeritics AutoChem II 2920 apparatus. 50 mg catalyst was loaded into a U-shaped quartz tube and pretreated at 550 °C for 1 h in He, then cooled to 50 °C in the inert atmosphere. At 50 °C, C$_3$H$_8$ or C$_3$H$_6$ was directly pulsed into the catalyst using He (30 mL min$^{-1}$) as the carrier gas. The weakly adsorbed C$_3$H$_8$ or C$_3$H$_6$ were removed by sweeping pure He at 50 °C for 1 h. Then the TPD measurement was conducted over the range 50–550 °C at a heating rate of 10 °C/min in a He flow. The temperature-programmed reactions were performed in the same packed-bed quartz tube as catalytic testing. The reaction gases (40 mL min$^{-1}$; "C$_2$H$_6$-C$_3$H$_6$": 20% C$_2$H$_6$, 6.25% C$_3$H$_6$, 20% O$_2$, and 53.75% N$_2$; "Single-C$_2$H$_6$": 20% C$_2$H$_6$, 20% O$_2$, and 60% N$_2$; "Single-C$_3$H$_6$": 6.25% C$_3$H$_6$, 20% O$_2$, and 73.75% N$_2$) were fed into the tube for 1 h at 30 °C to ensure uniform mixing, and then increased temperature at 2 °C min$^{-1}$. The effluents ($m/z$ of C$_2$H$_6$, C$_3$H$_6$, C$_2$H$_4$, and CO are 30, 39, 25, and 12, respectively) were measured via a mass spectrometer. Pulsed experiments were performed in the same packed-bed quartz tube as catalytic testing. Took the C$_3$D$_8$ pulse experiment as an example, C$_3$D$_8$ (l mL each time) was directly pulsed into the ODH reaction environment (40 mL min$^{-1}$; 6.25% C$_3$H$_6$, 20% O$_2$, and 73.75% N$_2$) at different temperatures. The products were analyzed by a mass spectrometer with the following mass-to-charge ($m/z$) signals: 34 for C$_3$D$_8$, 29 for C$_2$H$_4$, 58 for PO, and 64 for PO(D).

## Computational method

All calculated stationary points were optimized at B3LYP/6-31g(d, p) level[37–39] by Gaussian09 package[40]. Vibrational frequencies were calculated to identify the nature of the stationary points, either as minima or transition states, and abstract the thermodynamic data at 298.15 K and 1 atm. The intrinsic reaction coordinate (IRC) method[41] was used to confirm that each transition state connects the two minima along the reaction pathway.

## Data availability

The authors declare that all the relevant data within this paper and its Supplementary Information file are available from the corresponding authors upon a reasonable request. Source data are provided with this paper.

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

## Acknowledgements
This work was supported by state key program of National Natural Sci-
ence Foundation of China (21733002 to A.-H.L.), the National Key
Research and Development Program of China (2018YFA0209404 to A.-
H.L.), the Fundamental Research Funds for the Central Universities
(82233001 to Z.K.L.), the Young Scientists Fund of the National Natural
Science Foundation of China (22209018 to Z.Y.L.).

## Author contributions
A.-H.L. conceived the idea for the project. Z.K.L., J.F and B.G. conducted
the material synthesis. Z.K.L. and F.W. performed the structural char-
acterizations and catalytic tests. W.-D.L, J.S., and B.Q. discussed the
catalytic results. ZY.L. and D.Q.W. carried out the DFT simulations and
analysis. Z.K.L. and Z.Y.L drafted the manuscript under the guidance of
A.-H.L. and D.Q.W. All the authors discussed the results and commented
on the paper.

## Competing interests
The authors declare no competing interests.

## Additional information
**Supplementary information** The online version contains
supplementary material available at

An-Hui Lu.

**Peer review information** *Nature Communications* thanks Bert Wec-
khuysen, Melissa Cendejas and the other, anonymous, reviewer for their
contribution to the peer review of this work. Peer reviewer reports are
available.

