## [Peer Review File · Nature Communications]

Auto-accelerated dehydrogenation of alkane assisted by in-situ formed olefins over boron nitride under aerobic conditionsREVIEWER COMMENTS

Reviewer #1 (Remarks to the Author):

I have read this article with interest as it is important to activate light alkanes by processes such as ODH. The main aim of the study is to investigate the role of airborne olefins (the reaction product of alkane dehydrogenation) on the process of ODH. A lot of interesting experiments are performed but for me a basic question is not answered; what is the thermodynamically expected outcome when the olefins are present together with the alkanes. Hence, the question is if we are now really looking to a "reaction product"-promotion effect or not. I am not convinced that this is the case, and the experiments described do not provide for me sufficient evidence that this is indeed the case. Furthermore, the characterisation of the catalyst system can be performed in a more rigorous manner. It is not yet clear if the mechanistic aspects described are indeed taking place. Hence, I recommend either rejection of the paper or major revision in which the authors address my comments/criticism raised above.

Reviewer #2 (Remarks to the Author):

The manuscript entitled „Auto-accelerated dehydrogenation of alkane assisted by online-born olefins over boron nitride under aerobic conditions “ reports on a positive effect of in situ formed olefins on the rate of alkane conversion in the oxidative dehydrogenation of ethane or propane. Several catalytic tests have been carried out, while no sophisticated catalyst characterization studies have been conducted. Some theoretical calculations have also been carried out to assist in understanding of the mechanism of the reaction. This study is original, although the role of gas-phase radicals in the oxidative conversion of C1-C4 alkanes has already been reported in several studies, e.g. [doi.org/10.1016/S0021-9517\(02\)00116-1](https://doi.org/10.1016/S0021-9517(02)00116-1), doi.org/10.1038/s42004-020-00445-3, doi.org/10.1002/anie.202002440, doi.org/10.1002/anie.202003695. The present authors introduce an alternative mechanistic concept, which “... offered a method for efficiently producing olefins”. I feel that this quotation from the abstract is speculative and overestimates the impact of this study. It is not surprising that higher alkane or olefins can be activated at lower temperature than their lower counterparts. What is interesting is that the conversion of the latter can be enhanced by co-fed higher alkane or olefins probably due to the formation of radicals promoting the conversion. However, no improvements in the selectivity to the target olefins can be achieved. The authors briefly refer to Figure S14 showing high olefin selectivity in C₂H₆-C₃H₈ tests without, however specifying the selectivity to C₂H₄ and C₃H₆. The degree of C₂H₆ and C₃H₈ conversion is also not defined. In contrast to Figure S14, Figure S15 shows that the selectivity to C₂H₄ is worsened when dehydrogenating C₂H₆ in the presence of C₃H₈. It is also unclear from an industrial viewpoint, if it is economically feasible to accelerate C₂H₆ dehydrogenation by C₃H₈ without a significant improvement in the selectivity to C₂H₄. The selectivity to C₃H₆ through C₃H₈ dehydrogenation in this case is also not improved.

Apart from the above general comments, the below specific comments and suggestions should be addressed.

i) It is unclear why propane conversion should depend on how the catalyst is tested, i.e. in one reactor

or in two reactors connected in a series. The degree of propane conversion according to Figure 1A must be equal to that if only R1 is used but contains double catalyst amount (R1 + R2).

ii) Why do the authors conclude that previous studies excluded deep oxidation of olefins on BN catalysts. According to Ref. 7, the selectivity to olefins decreases with rising C₃H₈ conversion. The decrease is related to consecutive conversion of primarily formed C₃H₆.

iii) Figure 2B,C. How was possible to vary C₃H₈ conversion at a constant WHSV? If the authors changed the reaction temperature to vary the conversion, the obtained selectivity-conversion plots are not correct to draw conclusions about the scheme of product formation.

iv) It is not justified why propylene oxide is formed through a reaction between C₃H₈ and C₃H₆. How is this possible?

v) To prove that "... the synergistic effect of the two hydrocarbons occurred in the gas phase", blank tests without any catalyst are required. If gas-phase reactions played the major role, there would be no difference between BN, V-Al₂O₃ and Li-MgO.

vi) The positive role of propylene oxide for the selectivity to the desired olefins is questionable because this intermediate is preferentially oxidized to CO_x (Figure S8).

vii) I am wondering why the authors do not consider C_nH_{2n-1}O₂ radicals, which are typical intermediates in alkane oxidation reactions.

viii) The authors consider the reaction of OC₃H₇ radical with C₃H₆ but not with C₃H₈. The latter is present in significantly higher amounts. If OC₃H₇ reacts with C₃H₆, C₃H₇ radical must be formed. This radical can react with O₂ and C₃H₈ present in high amounts. What is about such possibilities?

ix) How OOH radical should be formed?

x) Figure 4 is incomplete and speculative.

Reviewer #3 (Remarks to the Author):

In this contribution, the authors aim to unravel the complex reaction mechanism responsible for the high product selectivity seen in the oxidative dehydrogenation over hBN. Using a combination of co-feeding and isotopic labelling experiments, the authors show that the alkene products (either co-fed or formed in situ) enhance the rate of alkane conversion. Using DFT-calculated reaction barriers, the authors propose an expanded gas-phase reaction mechanism which includes the role of propylene and transient propylene oxide species. While the results contained in this manuscript are interesting and can provide valuable insight into this complex and ever-evolving reaction mechanism, the manuscript in its current form is missing many critical details and control experiments, the omission of which decreases the potential impact of this work and makes the manuscript difficult to understand and interpret. I suggest major revisions before considering publication.

1. The title is confusing and does not clearly convey what the subject of the paper. "Online-born" is a confusing phrase. I suggest "in situ formed," which might be more readily understood by the community.

2. Include full reaction details in all captions where relevant. Especially for the hydrocarbon flow rates,

as they are only stated once early in the manuscript, so I had to keep returning to that spot in the text to remind myself what the hydrocarbon flow rate was.

3. Figure 1:

a. Please specify how the conversion for “R2” was calculated.

b. Visually, conversion from $R1 + R2 > \text{Total conversion}$. Please put values on the bar graph to prevent confusion.

4. It would be helpful for comparison to include the single-alkane conversion in Table S1. Include hydrocarbon flow rates in table caption.

5. Please comment on your use of an undiluted hBN catalyst bed. You argue that the role of the olefin is to modify species in the gas phase. Venegas, et al. (Org. Process Res. Dev. 2018, 22, 12, 1644–1652), showed that the rate of propane consumption has a volcano-like dependence on the ratio of catalyst to diluent given the same bed volume, where having no diluent in the bed hinders the reaction rate, as hBN can quench radical species. I would be interested in seeing your system’s response when the hBN is diluted. As the addition of olefin enhances the alkane conversion rate through gas phase chemistry, I suspect that the enhancement effect would be even greater if the bed configuration was optimized to promote gas-phase chemistry.

6. Figure 2 is blurry and difficult to read.

7. Regarding the discussion of Figures 2A, S2, and S3, you state, “These results indicated that the activation temperature of reactants could be significantly decreased by co-feeding olefin with alkane.” (pp. 7, lines 97-98) However, I do not believe your analysis to be so straightforward. The three reactions you compare all have different HC:O₂ ratios and different total HC (hydrocarbon) contact times. I understand that you’re trying to keep specific HC contact times consistent, but in doing so, you change the overall gas composition, which complicates the interpretation of your results. By changing the HC:O₂ ratio, you change the amount of HC species that can participate in the gas-phase reaction, so it follows that the single-HC experiments, which have lower [HC] than in the alkane-olefin mixes, would exhibit lower reactivity and conversions because the total concentration of species that could participate in the reaction is lower (in A-B mode, 26.3 % of the feed is HC, while in A mode, 20% of feed is HC, and in B mode, 6.3 % of the feed is HC). So rather than probing the synergistic effect of adding an olefin, you partially probe the effect of changing the HC:O₂ ratio. I suggest an additional set of experiments where the HC:O₂ ratio is kept consistent between the different reaction feeds to show that the increased reactivity is indeed from the synergistic effect and not an effect of changing the HC concentration in the gas feed.

a. These issues of changing [HC] persist throughout the entire manuscript.

8. Pp.8, line 118, “...indicating more consumption of propane at longer duration.” “duration” should be changed to “contact time” to be more precise.

9. What is the reaction temperature for Figures 2B, C, and S5B?

10. How do the product distributions compare between the single-alkane and alkane-olefin feeds?

a. How do the alkane-olefin feed conversion and product selectivity respond to changes in WHSV?

11. Figure 2D is incredibly blurry and difficult to read.

a. What was happening with C₃H₆ during the pulse experiments? Given the hypothesis that a “co-reaction of the two hydrocarbons” is responsible for the observed C-C cleavage, C₃H₆ should also be consumed. We should see a similar consumption to what is shown in Figure S7 for ethylene.

- b. Along the same lines, what does the production of C₃D₆ look like?
- c. Was there any time delay in dosing and product formation? Especially given the hypothesis that PO(D) is a secondary product. There is no time scale on the x-axis, so I cannot tell for myself.
12. Figure S7 is incredibly difficult to read.
13. What are the reaction conditions used in Figure S8?
14. On pp. 13, lines 202-203, you state, "According to our experimental measurement, co-feeding of propylene results in a moderate increase of CO formation." Where is this data shown? Please show the product distributions for the reactions you are comparing.
15. Pp. 14, line 220, "evolvment" should be "evolution"
16. Figure 4 is not at all clear:
- Which H is abstracted in the initial "H-abstraction" step? Given the different bond strengths of primary and secondary C-H bonds, I would expect different free energy barriers.
 - Similarly, which C is the O bonding to? Does it make a difference?
 - This proposed reaction network is not in line with the product distributions observed for propane ODH
- This proposed reaction network does not explain product distributions that have been previously reported in the literature. For example, according to Figure 4, C₂H₄ and CO are produced in a 1:1 ratio. Multiple studies report a higher ratio, indicating that there must be another route to ethylene formation that is not accounted for in the proposed network.
 - The proposed reaction network suggests that the only route to ethylene is the degradation of PO. If PO is pulsed into the reactor as done with other species in Fig. 2, do you observe an increase in ethylene?
 - In the degradation of C₃H₆O  C₂H₄ + CO, what happens to the other 2 Hs?
17. Figure 5 suffers from the same comparison issues that I noted earlier, as total [HC] differs pretty severely between the single-feed and co-feed experiments. Too many important variables change between experiments for the comparison to be meaningful.
- Again, missing reaction details in figure caption.
18. Why are Figures 5E and 5F run at different temperatures?
19. Pp. 18 lines 278-280, "Remarkably, the added propane may not lead to irreversible positive or negative effects over BN catalyst because similar ethane conversions were obtained under three cycles of "Single-C₂H₆"." This observation is in line with Venegas, et al., who saw a similar promotional effect when co-feeding water in propane ODH over hBN (Angew. Chem. Int. Ed. 2020, 59, 16527.), which they similarly attributed to contributions in the gas phase, as water can enhance the radical pool concentration. It seems the added olefin is modifying the reaction similarly, although without the full product distributions, it is hard to say whether the addition of olefin enhances existing reaction pathways, or opens up additional reaction pathways. I'm curious to know what the product distributions obtained in the experiments here are.

Overall, I think these results are very interesting and I look forward to seeing the improved manuscript! It's a truly fascinating system and your observation of PO formation is an important insight.

-Dr. Melissa Cendejas

Response to the reviewers' comments

Reviewer #1

I have read this article with interest as it is important to activate light alkanes by processes such as ODH. The main aim of the study is to investigate the role of airborne olefins (the reaction product of alkane dehydrogenation) on the process of ODH. A lot of interesting experiments are performed but for me a basic question is not answered; what is the thermodynamically expected outcome when the olefins are present together with the alkanes. Hence, the question is if we are now really looking to a "reaction product"-promotion effect or not. I am not convinced that this is the case, and the experiments described do not provide for me sufficient evidence that this is indeed the case. Furthermore, the characterisation of the catalyst system can be performed in a more rigorous manner. It is not yet clear if the mechanistic aspects described are indeed taking place. Hence, I recommend either rejection of the paper or major revision in which the authors address my comments/criticism raised above.

Reply: We acknowledge the reviewer for the positive recommendation and constructive comments on our work. The reviewer mainly raised questions about the thermodynamically expected outcome in an olefin-existent environment and whether a "reaction product"-promotion effect was actually observed in the ODH reaction. In addition, the reviewer suggested rigorously performing some characterizations of catalysts. Hence, we would answer the above questions in three parts as follows.

1. The ODH reaction of propane is a thermodynamically unrestricted process. According to the established understanding in previous studies from our group and other groups, the ODH reactions of light alkanes follow a mechanism with the co-presence and cooperation of surface-mediated radical formation processes and gas phase channels. The coupling of the two types of channels results in the remarkable performance of boron-based catalysts in the ODH of light alkanes with high selectivity for olefin products and low yield of over-oxidation products. The radical-mediated mechanism implicates the co-presence of species that can be easily activated, here the co-fed olefins, may enhance the breeding of reactive radical species in the reaction system. According to our previous studies (*J. Phys. Chem. Lett.*, 2021, 3, 8770-8776), the boron sites have a unique ability to eliminate alkoxy radicals by which the pathway to deep oxidation is blocked, thus the rich generation of radical species is expected to enhance the conversion of alkanes, but do not harm

much the selectivity for olefin products. Thus, based on the established knowledge of the unique mechanism of BN and the thermodynamic properties of the ODH reaction, we expect that co-feeding of olefin could interact with some radicals to highly active species and further promote alkane activation. This expectation is consistent with our observations in the current experimental study.

2. To elucidate the promotion role of in situ generated olefin, a tandem reaction system (Figure R1) was adopted to decompose the catalyst bed into two parts in which the conversions of propane in the upper catalyst may be measured and the gas flow with the co-existence of nascent olefin products and the unreacted propane was fed in the lower catalyst.

As shown in Figure R1, the composition of “Reactor 2” inlet gas was the same as that of “Reactor 1” exhaust, which contained the residual reactant (C_3H_8 and O_2) and the ODH products (C_3H_6 , C_2H_4 , CO , and CO_2). The measured conversion of propane in “Reactor 2” was higher than that in “Reactor 1” (Reactor 1: 9.1%, Reactor 2: 12.8%), evidencing that the “reaction product”-promotion effect did exist in the reaction system. We added the explanation in page 4, line 62.

Figure R1 The design of the tandem reaction experiment to identify the promotion role of olefin products.

For comparison, we simulated “Reactor 2” inlet gas with and without added propylene for oxidative dehydrogenation (Supplementary Fig. 2). When propylene was absent, the propane conversion was 7.2%. However, the added propylene promoted propane conversion to 11.7% under the same ODH conditions, which was close to the conversion in “Reactor 2” of the tandem reaction

system. These outcomes indicated that the propane conversion in “Reactor 2” can be promoted by the in situ formed olefins in “Reactor 1”. We added the corresponding discussion in page 5, line 85.

Supplementary Fig. 2 C₃H₈ conversion under the simulated R2 inlet gas with and without added C₃H₆.

We also conducted a “Single-C₃H₈ (10.5 sccm)” experiment where the molar ratio of hydrocarbon to oxygen (HC: O₂) was the same as that in the “C₃H₈-C₃H₆” mode. As shown in Supplementary Fig. 3, the C₃H₈ conversion was lower than that in the “C₃H₈-C₃H₆” mode and slightly higher than that in the “Single-C₃H₈” mode reported in the previous version of the manuscript. The difference in the C₃H₈ conversion between the two feeding modes implies that the increased conversion of C₃H₈ in the “C₃H₈-C₃H₆” mode was the consequence of the addition of olefin. A brief discussion was added in page 6, line 92, to address this issue.

Supplementary Fig. 3 C₃H₈ conversion in “Single-C₃H₈ (8 sccm)”, “Single-C₃H₈ (10.5 sccm)” and “C₃H₈-C₃H₆” modes. F_{total} = 40 mL min⁻¹. “Single-C₃H₈ (8 sccm)”, C₃H₈: O₂: N₂ = 8: 8: 24; “Single-C₃H₈ (10.5 sccm)”, C₃H₈: O₂: N₂ = 10.5: 8: 21.5; “C₃H₈-C₃H₆”, C₃H₈: C₃H₆: O₂: N₂ = 8: 2.5: 8: 21.5.

In addition, in the previous study, the signal of olefins was detected a few centimeters above the surface of BN catalysts (*Angew. Chem. Int. Ed.*, 2020, 59, 8042-8046), indicating the co-existence of alkane and olefins in the atmosphere under ODH conditions. Together with this observation, it is reasonable for us to conceive the occurrence of the synergy between alkane and olefins in propane ODH reaction.

3. Following the suggestions of the reviewer, a series of characterizations were conducted to capture the state of catalysts after ODH reactions under different atmospheres. In the previous version of the manuscript, the synergistic interaction between alkanes and alkenes was considered to occur mainly in the gas phase rather than on the catalyst surface, thus the catalyst was not systematically characterized.

In these characterizations, four catalysts were considered, respectively the catalysts at different stages of the ODH reactions: fresh BN, activated BN, BN after reaction in “Single-C₃H₈” mode for 3 h (Single-C₃H₈-BN), and BN after reaction in “C₃H₈-C₃H₆” mode for 3 h (C₃H₈-C₃H₆-BN). The specific treatment processes for catalysts were shown in **Supplementary Table 2**. In the FT-IR spectra, the band at around 3400 cm⁻¹ was assigned to O-H vibration that appeared on the activated BN (**Supplementary Fig. 4**). The X-ray diffraction (XRD) pattern of activated BN exhibited a new diffraction peak at $2\theta = 14.6^\circ$ corresponding to the generation of BO_x (**Supplementary Fig. 5**). In the NH₃-TPD experiment, activated BN showed a new NH₃ desorption peak, which also proved the existence of BO_x (**Supplementary Fig. 6**). SEM showed that the boron nitride sheets changed from uniform dispersion of fresh BN to aggregation of activated BN (**Supplementary Fig. 7**). These outcomes indicated that active species formed on the catalyst surface under the ODH conditions.

When the activated BN were treated for another 3 h in the two atmospheres (“Single-C₃H₈”, “C₃H₈-C₃H₆”), the surface functional groups, acid content, and morphology were almost unchanged and no new species were regenerated.

The above results demonstrated that the activated BN was stable and would not be affected by the composition of the reaction gas. The corresponding discussion was added in page 7, line 104. And more detailed analyses were added in the supplementary information.

Supplementary Table 2 Specific treatments of the catalysts.

Catalysts	Treatments
fresh BN	-
activated BN	$F_{\text{total}} = 40 \text{ mL min}^{-1}$, $\text{C}_3\text{H}_8: \text{O}_2: \text{N}_2 = 8: 8: 24$, $550 \text{ }^\circ\text{C}$, 3 h
Single- C_3H_8 -BN	Process 1: $F_{\text{total}} = 40 \text{ mL min}^{-1}$, $\text{C}_3\text{H}_8: \text{O}_2: \text{N}_2 = 8: 8: 24$, $550 \text{ }^\circ\text{C}$, 3 h Process 2: $F_{\text{total}} = 40 \text{ mL min}^{-1}$, $\text{C}_3\text{H}_8: \text{O}_2: \text{N}_2 = 8: 8: 24$, $500 \text{ }^\circ\text{C}$, 3 h
C_3H_8 - C_3H_6 -BN	Process 1: $F_{\text{total}} = 40 \text{ mL min}^{-1}$, $\text{C}_3\text{H}_8: \text{O}_2: \text{N}_2 = 8: 8: 24$, $550 \text{ }^\circ\text{C}$, 3 h Process 2: $F_{\text{total}} = 40 \text{ mL min}^{-1}$, $\text{C}_3\text{H}_8: \text{C}_3\text{H}_6: \text{O}_2: \text{N}_2 = 8: 2.5: 8: 24$, $500 \text{ }^\circ\text{C}$, 3 h

Supplementary Fig. 4 FT-IR spectra of the fresh BN, activated BN, spent BN in the Single- C_3H_8 mode (Single- C_3H_8 -BN), and spent BN in the C_3H_8 - C_3H_6 mode (C_3H_8 - C_3H_6 -BN).

Supplementary Fig. 5 XRD patterns of the fresh BN, activated BN, spent BN in the Single-C₃H₈ mode (Single-C₃H₈-BN), and spent BN in the C₃H₈-C₃H₆ mode (C₃H₈-C₃H₆-BN).

Supplementary Fig. 6 NH₃-TPD results of the fresh BN, activated BN, spent BN in the Single-C₃H₈ mode (Single-C₃H₈-BN), and spent BN in the C₃H₈-C₃H₆ mode (C₃H₈-C₃H₆-BN).

Supplementary Fig. 7 SEM images of the fresh BN, activated BN, spent BN in the Single-C₃H₈ mode (Single-C₃H₈-BN), and spent BN in the C₃H₈-C₃H₆ mode (C₃H₈-C₃H₆-BN).

Reviewer #2

The manuscript entitled “Auto-accelerated dehydrogenation of alkane assisted by online-born olefins over boron nitride under aerobic conditions” reports on a positive effect of in situ formed olefins on the rate of alkane conversion in the oxidative dehydrogenation of ethane or propane. Several catalytic tests have been carried out, while no sophisticated catalyst characterization studies have been conducted. Some theoretical calculations have also been carried out to assist in understanding of the mechanism of the reaction. This study is original, although the role of gas-phase radicals in the oxidative conversion of C₁-C₄ alkanes has already been reported in several studies, e.g. [doi.org/10.1016/S0021-9517\(02\)00116-1](https://doi.org/10.1016/S0021-9517(02)00116-1), doi.org/10.1038/s42004-020-00445-3, doi.org/10.1002/anie.202002440, doi.org/10.1002/anie.202003695. The present authors introduce an alternative mechanistic concept, which “... offered a method for efficiently producing olefins”.

I feel that this quotation from the abstract is speculative and overestimates the impact of this study. It is not surprising that higher alkane or olefins can be activated at lower temperature than their lower counterparts. What interesting is that the conversion of the latter can be enhanced by co-fed higher alkane or olefins probably due to the formation of radicals promoting the conversion. However, no improvements in the selectivity to the target olefins can be achieved. The authors briefly refer to Supplementary Fig. 14 showing high olefin selectivity in C₂H₆-C₃H₈ tests without, however specifying the selectivity to C₂H₄ and C₃H₆. The degree of C₂H₆ and C₃H₈ conversion is also no defined. In contrast to Supplementary Fig. 14, Supplementary Fig. 15 shows that the selectivity to C₂H₄ is worsen when dehydrogenating C₂H₆ in the presence of C₃H₈. It is also unclear from an industrial viewpoint, if it is economically feasible to accelerate C₂H₆ dehydrogenation by C₃H₈ without a significant improvement in the selectivity to C₂H₄. The selectivity to C₃H₆ though C₃H₈ dehydrogenation in this case is also not improved.

Reply: We acknowledge the reviewer for grading our work that “this study is original”. The reviewer mainly raised concerns about the selectivity of olefins in co-feeding mode and the impact of this study. In addition, some characterizations were suggested to be performed. Below we address these concerns in three aspects:

1. Following the suggestions of the reviewer, a series of characterizations were conducted to assess the state of catalysts after ODH reactions under different atmospheres. In the previous version of the manuscript, since the synergistic interaction between alkanes and alkenes was considered to mainly occur in the gas phase rather than on the catalyst surface, these systematic characterizations of the catalyst were not reported.

In the characterizations, four catalysts were considered: fresh BN, activated BN, BN after reaction in “Single-C₃H₈” mode for 3 h (Single-C₃H₈-BN), and BN after reaction in “C₃H₈-C₃H₆” mode for 3 h (C₃H₈-C₃H₆-BN). The specific treatment processes for catalysts were shown in **Supplementary Table 2**. The FT-IR spectra showed the band at around 3400 cm⁻¹ assigned to O-H vibration appeared on the activated BN (**Supplementary Fig. 4**). The X-ray diffraction (XRD) pattern of activated BN exhibited a new diffraction peak at $2\theta = 14.6^\circ$ corresponding to the generation of BO_x (**Supplementary Fig. 5**). In the NH₃-TPD experiment, activated BN showed a new NH₃ desorption peak, which also proved the existence of BO_x (**Supplementary Fig. 6**). SEM showed that the boron nitride sheets changed from uniform dispersion of fresh BN to aggregation

of activated BN (Supplementary Fig. 7). These outcomes indicated that the active species were formed under the ODH reaction atmosphere. When the activated BN were treated for another 3 h under the two atmospheres (“Single-C₃H₈”, “C₃H₈-C₃H₆”), the surface functional groups, acid content, and morphology were almost unchanged and no new species were regenerated.

The above results demonstrated that the activated BN was stable and not affected by the composition of the reaction gas. The corresponding discussion was added in page 7, line 104. And more detailed analyses were added in the supplementary information.

Supplementary Table 2 Specific treatment processes for the catalysts.

Catalysts	Treatments
fresh BN	-
activated BN	$F_{\text{total}} = 40 \text{ mL min}^{-1}$, C ₃ H ₈ : O ₂ : N ₂ = 8: 8: 24, 550 °C, 3 h
Single-C ₃ H ₈ -BN	Process 1: $F_{\text{total}} = 40 \text{ mL min}^{-1}$, C ₃ H ₈ : O ₂ : N ₂ = 8: 8: 24, 550 °C, 3 h Process 2: $F_{\text{total}} = 40 \text{ mL min}^{-1}$, C ₃ H ₈ : O ₂ : N ₂ = 8: 8: 24, 500 °C, 3 h
C ₃ H ₈ -C ₃ H ₆ -BN	Process 1: $F_{\text{total}} = 40 \text{ mL min}^{-1}$, C ₃ H ₈ : O ₂ : N ₂ = 8: 8: 24, 550 °C, 3 h Process 2: $F_{\text{total}} = 40 \text{ mL min}^{-1}$, C ₃ H ₈ : C ₃ H ₆ : O ₂ : N ₂ = 8: 2.5: 8: 24, 500 °C, 3 h

Supplementary Fig. 4 FT-IR spectra of the fresh BN, activated BN, spent BN in the Single-C₃H₈ mode (Single-C₃H₈-BN), and spent BN in the C₃H₈-C₃H₆ mode (C₃H₈-C₃H₆-BN).

Supplementary Fig. 5 XRD patterns of the fresh BN, activated BN, spent BN in the Single-C₃H₈ mode (Single-C₃H₈-BN), and spent BN in the C₃H₈-C₃H₆ mode (C₃H₈-C₃H₆-BN).

Supplementary Fig. 6 NH₃-TPD results of the fresh BN, activated BN, spent BN in the Single-C₃H₈ mode (Single-C₃H₈-BN), and spent BN in the C₃H₈-C₃H₆ mode (C₃H₈-C₃H₆-BN).

Supplementary Fig. 7 SEM images of the fresh BN, activated BN, spent BN in the Single-C₃H₈ mode (Single-C₃H₈-BN), and spent BN in the C₃H₈-C₃H₆ mode (C₃H₈-C₃H₆-BN).

2. In the abstract, the phrase “offered a method for” was changed to “provided an insight into” in page 2, line 21.

3. Regarding the selectivity: in the “C₂H₆-C₃H₈” mode, the product C₃H₆ from C₃H₈ would occupy part of selectivity in the calculation of product distribution, leading to a seemingly lowered selectivity for C₂H₄. To precisely evaluate the selectivity in a fair way, we compared the products selectivity at similar conversions of alkane in the two feeding modes (mode 1: “C₂H₆-C₃H₈”; mode 2: “Single-C₂H₆” + “Single-C₃H₈”). As shown in Supplementary Table 3, the selectivity for olefins in the co-feeding mode was essentially the same as that in the single-feeding mode. This shows the advantage of the co-feeding mode that high conversion can be achieved at lower reaction temperatures without significantly changing the selectivity of olefins.

Supplementary Table 3 Products distribution at similar alkane conversions under different feeding modes.

Mode	Conversion (%)			Selectivity (%)			
	C ₂ H ₆	C ₃ H ₈	Alkanes	C ₂ H ₄	C ₃ H ₆	CO	Olefins
C ₂ H ₆ -C ₃ H ₈ (510 °C)	4.0	8.0	5.2	57.0	42.6	0.4	99.6
Single- C ₂ H ₆ (540 °C)	3.9		3.9	100.0			100.0
Single- C ₃ H ₈ (530 °C)		8.5	8.5	10.4	89.0	0.6	99.4
“Single- C ₂ H ₆ ” + “Single- C ₂ H ₈ ”			5.3	54.4	45.3	0.3	99.7

We added the specific alkane conversion and olefin selectivity in Supplementary Fig. 24 and Supplementary Table 3 and added a brief discussion on page 19, line 302 to address this issue.

Supplementary Fig. 24 Alkane conversions and product selectivity in “C₂H₆-C₃H₈” mode as a function of temperature over BN. F_{total} = 40 mL min⁻¹, C₂H₆: C₃H₈: O₂: N₂ = 8: 2.5: 8: 21.5.

Apart the above general comments, the below specific comments and suggestions should be addressed.

i) It is unclear why propane conversion should depend on how the catalyst is tested, i.e. in one reactor or in two reactors connected in a series. The degree of propane conversion according to Figure 1A must be equal to that if only R1 is used but contains double catalyst amount (R1 + R2).

Reply: The purpose of designing a tandem reaction system is to evaluate the promotion effect of in-situ formed propylene on the conversion of residual propane by decomposing the catalyst bed into two parts to detect the conversions of propane in the upper catalyst and lower catalyst, respectively.

As shown in **Figure R1**, the exhaust of “Reactor 1”, which contained residual reactants (C_3H_8 and O_2) and the in-situ formed products, was fed into “Reactor 2”. The measured conversion of propane in “Reactor 2” was higher than that in “Reactor 1” (Reactor 1: 9.1%, Reactor 2: 12.8%), evidencing that the “reaction product”-promotion effect did exist in the reaction system. We added the explanation in **page 4, line 62**.

Figure R1 The design of a tandem reaction experiment.

Owing to the complexity of the reaction system, which involves gas phase radical reactions and quenching, the conversion of propane does not linearly depend on catalysts loading (*Org. Process Res. Dev.*, 2018, 22, 1644-1652, Figure 8; *Angew. Chem. Int. Ed.*, 2021, 60, 19691-19695, Supplementary Fig. 7). Therefore, the degree of propane conversion may not be equal to that obtained in the experiment with only R1 used containing the same amount of catalyst (**R1 + R2, Figure R1, left column**).

ii) Why do the authors conclude that previous studies excluded deep oxidation of olefins on BN catalysts. According to Ref. 7, the selectivity to olefins decreases with rising C_3H_8 conversion. The decrease is related to consecutive conversion of primarily formed C_3H_6 .

Reply: In previous study propylene feeding in the reactor under the same conditions as ODH reaction was found not to be oxidized, leading to a conclusion that “there is no direct over-oxidation of olefins over the h-BN catalyst” (*Sci. Adv.*, 2019, 5, eaav8063, Page 3, line 25). This suggests that in the absence of alkane, propylene is not activated under the conditions the same as the ODH reaction. The statement in **page 5, line 80** was modified to avoid confusion.

In our present study, the tandem design of the reaction system showed that the Reactor 1 exhaust contained both propane and propylene. The feeding of this mixture with the co-presence of propane and propylene to Reactor 2 caused a significantly higher conversion of propane, implicating that co-presence of propane and propylene can activate propylene which is responsible for the increased conversion of propane. This thus offered direct evidence of the existence of propylene oxidative in the propane ODH reaction system.

In previous studies, higher C_3H_8 conversion was generally achieved by increasing temperature. This complicated the pathways in the ODH reactions, and to our knowledge, it has not been demonstrated whether the decrease of propylene selectivity is due to consecutive oxidation of propylene or the enhancement of other side-reaction pathways at higher temperatures before the formation of propylene. This is a different issue from our present work.

iii) Figure 2B, C. How was possible to vary C_3H_8 conversion at a constant WHSV? If the authors changed the reaction temperature to vary the conversion, the obtained selectivity-conversion plots are not correct to draw conclusions about the scheme of product formation.

Reply: We agree with the reviewer that it is hard to correlate selectivity with conversion obtained at different temperatures. Here in Figure 2B, we only want to prove that at constant conversion, the influence of WHSV on the product distribution is not an accidental phenomenon, and we didn't try to draw other conclusions from the selectivity-conversion plots to avoid over-interpretation of the data.

iv) It is not justified why propylene oxide is formed through a reaction between C_3H_8 and C_3H_6 . How is this possible?

Reply: In our pulse experiment, propylene oxide (PO) was detected and its amount increased when propylene and propane were co-fed. DFT calculations showed that it was feasible to produce PO via the reaction of propylene with alkoxy radicals and other oxidative radicals, e.g. peroxy, under ODH conditions given the calculated reaction energy barriers.

v) To prove that "... the synergistic effect of the two hydrocarbons occurred in the gas phase", blank tests without any catalyst are required. If gas-phase reactions played the major role, there would be no difference between BN, V- Al_2O_3 and Li-MgO.

Reply: Following the suggestion of the reviewer, we added the data obtained from blank tests without catalyst. At the temperature below 540°C, there was essentially no difference in the low

conversion of alkane whether the second alkane was present or not. Raising the temperature to a higher temperature resulted in a noticeable increase in the conversion of alkanes in the “C₂H₆-C₃H₈” mode compared to that in the “Single-C₂H₆” mode (Supplementary Fig. 30). Loading BN catalyst in the reactor, in contrast, resulted in a nearly 5-fold higher conversion of ethane under the co-feeding atmosphere than that under the ethane single-feed atmosphere at 530 °C. These results demonstrated the important role of BN in decreasing the reaction temperature and boosting the synergistic effect. In addition, BN showed a similar promoting effect in the “C₃H₈-C₃H₆” mode (Supplementary Fig. 19).

The reactions catalyzed by V-Al₂O₃ and Li-MgO follow different mechanisms and mainly occurred on the surface at tested temperature. It is thus hard for these two catalysts to breed the formation of more radicals required by a measurable synergistic effect between the two kinds of co-fed hydrocarbons. Brief discussions were added in page 20, line 318, page 20, line 328, and page 12, line 207.

Supplementary Fig. 30 C₂H₆ and C₃H₈ conversions as a function of temperature in the empty reactor. F_{total} = 40 mL min⁻¹, C₂H₆: C₃H₈: O₂: N₂ = 8: 2.5: 8: 21.5.

Supplementary Fig. 19 C₃H₈ and C₃H₆ conversions as a function of temperature in the empty reactor. $F_{\text{total}} = 40 \text{ mL min}^{-1}$, C₃H₈: C₃H₆: O₂: N₂ = 8: 2.5: 8: 21.5.

vi) The positive role of propylene oxide for the selectivity to the desired olefins is questionable because this intermediate is preferentially oxidized to CO_x (Supplementary Fig. 8).

Reply: We agree with the reviewer. In the propane ODH reaction, the selectivity of CO_x would increase when raising the temperature (Figure R2), indicating higher temperature benefited the deep-oxidized routes, which include the oxidation of PO. In the present study, the selectivity of olefins remained above 95% even when the propane conversion was over 20%, indicating the contribution of deep-oxidized routes was little.

Figure R2 C₃H₈ conversion and product selectivity in “Single-C₃H₈” mode as a function of temperature over BN. $F_{\text{total}} = 40 \text{ mL min}^{-1}$, C₃H₈: O₂: N₂ = 8: 8: 24.

vii) I am wondering why the authors do not consider $C_nH_{2n-1}O_2$ radicals, which are typical intermediates in alkane oxidation reactions.

Reply: We agree with the reviewer that $C_nH_{2n-1}O_2$ as a player in alkane oxidation reactions. The reason that we did not consider it here is in three aspects: (1) in the present work, instead of mapping the full reaction network, we only discussed the reaction pathways that are relevant to the promotion effect of additional propylene on the conversion of propane during propane ODH proposed by us (*J. Phys. Chem. C*, 2021, 125, 24930–24944) and other groups; (2) it is difficult to oxidize propylene directly in the boron nitride system (*Sci. Adv.*, 2019, 5, eaav8063). Thus, the activation of propane to $C_nH_{2n+1}O_2$ was given priority in this work; (3) the equilibrium constant for the reaction of the allyl radical, $\bullet C_3H_5 + O_2 \rightleftharpoons \bullet C_3H_5O_2$ was reported to be $\log(K_{eq})$ ($K_{eq} = [\bullet C_3H_5O_2] / [\bullet C_3H_5][O_2]$) = 5.0, 0.75, and -0.97 at 300, 500, and 600 K, respectively (*J. Am. Chem. Soc.*, 1965, 87, 972). The reaction would shift to the reverse side at increased temperature. Hence, $\bullet C_3H_5O_2$ was hard to be generated at the ODH reaction temperature.

viii) The authors consider the reaction of OC_3H_7 radical with C_3H_6 but not with C_3H_8 . The latter is present in significantly higher amounts. If OC_3H_7 reacts with C_3H_6 , the C_3H_7 radical must be formed. This radical can react with O_2 and C_3H_8 present in high amounts. What is about such possibilities?

Reply: It is feasible for $\bullet OC_3H_7$ radical to react with C_3H_8 . According to our calculations, this reaction is exergonic by 5.1 kcal/mol with a free energy barrier of 9.6 kcal/mol (please see **Table R1**). The reactions of $\bullet C_3H_7$ with O_2 in the gaseous channels have been reported in our previous work (Table 5 in *J. Phys. Chem. C*, 2021, 125, 24930–24944). The reaction of $\bullet C_3H_7$ with C_3H_8 can also happen facily once colliding with each other. Since the present work mainly focuses on the promotion effect of the co-fed olefin on the conversion of alkanes rather than a complete view of the reaction mechanisms, concerning that these above reaction pathways are less relevant to the phenomena observed in our experiments, we prefer not to discuss them in this work to avoid potential confusion.

Table R1 The key stationary points and relative free energies (ΔG^\ddagger and ΔG , in kcal/mol) in the propane activation by $C_3H_7O\bullet$ to form propyl radical in the gas phase

Eq.	Stationary points			Energies(kcal/mol)	
	RC	TS	IM	ΔG^\ddagger	ΔG

ix) How OOH radical should be formed?

Reply: There are mainly two formation ways: (1) reaction between boron species and oxygen on the surface of catalysts (*Angew. Chem. Int. Ed.*, 2020, 59, 16527-16535); (2) reaction between radicals and oxygen (*J. Phys. Chem. C*, 2021, 125, 24930–24944). Energetically these pathways are feasible under ODH conditions.

x) Figure 4 is incomplete and speculative.

Reply: Since this work focuses on the promotion effect of co-fed olefin on the conversion of alkane substrate under ODH conditions, in Figure 4 we only show the elementary steps we found to be relevant to the activation of propylene and its consumption rather than plotting a complex reaction network of the ODH reactions. For a relatively more complete reaction network, please see our recent work (*J. Phys. Chem. C*, 2021, 125, 24930–24944). As reviewer #3 said, this reaction system involved a complex and ever-evolving reaction mechanism. We are constantly making efforts to provide some valuable insight to improve the reaction network.

Reviewer #3

In this contribution, the authors aim to unravel the complex reaction mechanism responsible for the high product selectivity seen in the oxidative dehydrogenation over hBN. Using a combination of co-feeding and isotopic labelling experiments, the authors show that the alkene products (either co-fed or formed in situ) enhance the rate of alkane conversion. Using DFT-calculated reaction barriers, the authors propose an expanded gas-phase reaction mechanism which includes the role of propylene and transient propylene oxide species. While the results contained in this manuscript are interesting and can provide valuable insight into this complex and ever-evolving reaction mechanism, the manuscript in its current form is missing many critical details and control experiments, the omission of which decreases the potential impact of this work and makes the

manuscript difficult to understand and interpret. I suggest major revisions before considering publication.

Reply: Thank the reviewer for the positive recommendation.

1. The title is confusing and does not clearly convey what the subject of the paper. “Online-born” is a confusing phrase. I suggest “in situ formed,” which might be more readily understood by the community.

Reply: Following the suggestion of the reviewer, “Online-born” was replaced by “in-situ formed” in the title (page 1, line 1).

2. Include full reaction details in all captions where relevant. Especially for the hydrocarbon flow rates, as they are only stated once early in the manuscript, so I had to keep returning to that spot in the text to remind myself what the hydrocarbon flow rate was.

Reply: Thanks for the reviewer's reminder. We added reaction details in relevant captions.

3. Figure 1:

a. Please specify how the conversion for “R2” was calculated.

b. Visually, conversion from R1 + R2 > Total conversion. Please put values on the bar graph to prevent confusion.

Reply:

a. In the experiment, the R1 exhaust at 500 °C ($n_{C_3H_8(R1)}^{out, 500\text{ }^\circ\text{C}}$) as the feeding of R2 ($n_{C_3H_8(R2)}^{in}$). The method to calculate propane conversion in R2 is as follows, which is added in page 8 of supplementary information.

$$X_{C_3H_8(R2)} = \left(1 - \frac{n_{R2}^{out}}{EF * n_{C_3H_8(R2)}^{in}}\right) * 100 = \left(1 - \frac{n_{R2}^{out}}{EF * n_{C_3H_8(R1)}^{out, 500\text{ }^\circ\text{C}}}\right) * 100$$

b. According to the reviewer’s suggestion, we have put values on the bar graph.

4. It would be helpful for comparison to include the single-alkane conversion in Supplementary Table 1. Include hydrocarbon flow rates in table caption.

Reply: According to the reviewer's suggestion, we supplemented the above data in Supplementary Table 1.

5. Please comment on your use of an undiluted hBN catalyst bed. You argue that the role of the olefin is to modify species in the gas phase. Venegas, et al. (Org. Process Res. Dev. 2018, 22, 12, 1644–1652), showed that the rate of propane consumption has a volcano-like dependence on the ratio of catalyst to diluent given the same bed volume, where having no diluent in the bed hinders

the reaction rate, as hBN can quench radical species. I would be interested in seeing your system's response when the hBN is diluted. As the addition of olefin enhances the alkane conversion rate through gas phase chemistry, I suspect that the enhancement effect would be even greater if the bed configuration was optimized to promote gas-phase chemistry.

Reply: According to the reviewer's suggestion, we conducted a “C₃H₈-C₃H₆” ODH experiment with a diluted BN catalyst bed. As shown in Figure R3, propane conversion of diluted BN was slightly higher than that of undiluted BN, demonstrating that the reaction was enhanced by additional gas phase space. However, the promotion effect was not so obvious as in the work of Venegas, et al. This may be due to different test conditions.

Figure R3 C₃H₈ conversion in “C₃H₈-C₃H₆” mode catalyzed by undiluted and diluted BN as a function of temperature. F_{total} = 40 mL min⁻¹, C₃H₈: C₃H₆: O₂: N₂ = 8: 2.5: 8: 21.5. Diluent: 40-60 mesh SiO₂. Bed height, undiluted BN: 0.5 cm, undiluted BN: 1 cm.

6. Figure 2 is blurry and difficult to read.

Reply: We have enlarged the font and thickened the line to improve the readability of Figure 2.

7. Regarding the discussion of Figures 2A, S2, and S3, you state, “These results indicated that the activation temperature of reactants could be significantly decreased by co-feeding olefin with alkane.” (pp. 7, lines 97-98) However, I do not believe your analysis to be so straightforward. The three reactions you compare all have different HC:O₂ ratios and different total HC (hydrocarbon) contact times. I understand that you're trying to keep specific HC contact times consistent, but in

doing so, you change the overall gas composition, which complicates the interpretation of your results. By changing the HC:O₂ ratio, you change the amount of HC species that can participate in the gas-phase reaction, so it follows that the single-HC experiments, which have lower [HC] than in the alkane-olefin mixes, would exhibit lower reactivity and conversions because the total concentration of species that could participate in the reaction is lower (in A-B mode, 26.3 % of the feed is HC, while in A mode, 20% of feed is HC, and in B mode, 6.3 % of the feed is HC). So rather than probing the synergistic effect of adding an olefin, you partially probe the effect of changing the HC:O₂ ratio. I suggest an additional set of experiments where the HC:O₂ ratio is kept consistent between the different reaction feeds to show that the increased reactivity is indeed from the synergistic effect and not an effect of changing the HC concentration in the gas feed.

a. These issues of changing [HC] persist throughout the entire manuscript.

Reply: We agree with the reviewer and conducted “Single-C₃H₈ (10.5 sccm)” experiment where the HC: O₂ ratio was the same as that in “C₃H₈-C₃H₆” mode. As shown in Supplementary Fig. 3, the C₃H₈ conversion was lower than that in the “C₃H₈-C₃H₆” mode and slightly higher than that in the “Single-C₃H₈” mode of the first version, indicating that the increased reactivity of C₃H₈ in “C₃H₈-C₃H₆” mode was due to the addition of olefin rather than changing the HC: O₂ ratio. In the first submitted version, we could also observe the characteristic of olefin by comparing ethane conversion in “C₂H₆-C₃H₈” and “C₂H₆-C₃H₆” modes with the same HC: O₂ ratio. We added the corresponding discussion in page 6, line 92.

Supplementary Fig. 3 C₃H₈ conversion in “Single-C₃H₈ (8 sccm)”, “Single-C₃H₈ (10.5 sccm)” and “C₃H₈-C₃H₆” modes. F_{total} = 40 mL min⁻¹. “Single-C₃H₈ (8 sccm)”, C₃H₈: O₂: N₂ = 8: 8: 24; “Single-C₃H₈ (10.5 sccm)”, C₃H₈: O₂: N₂ = 10.5: 8: 21.5; “C₃H₈-C₃H₆”, C₃H₈: C₃H₆: O₂: N₂ = 8: 2.5: 8: 21.5.

8. Pp.8, line 118, “...indicating more consumption of propane at longer duration.” “duration” should be changed to “contact time” to be more precise.

Reply: Thank the reviewer for pointing it out. We have revised it, and the word “duration” was changed to “contact time” in page 9, line 150 as suggested.

9. What is the reaction temperature for Figures 2B, C, and S5B?

Reply: The reaction temperatures for Figures 2B, C, and S5B were 410 °C to 510 °C. The information was added in the figure caption.

10. How do the product distributions compare between the single-alkane and alkane-olefin feeds?

a. How do the alkane-olefin feed conversion and product selectivity respond to changes in WHSV?

Reply: We added “C₃H₈-C₃H₆” experiments in different WHSV (Supplementary Fig. 12). Because we could not know the consumption of co-fed propylene and the production of propylene formed from propane in the “C₃H₈-C₃H₆” mode, the product distribution could only be obtained by calculating the net production of C₃H₆, where the conversion of co-fed propylene was assumed to be 0. This method offers a lower limit of the calculated propylene selectivity because it doesn't distinguish the in situ formed propane-derived propylene and the consumed co-fed propylene. Furthermore, ethylene and CO_x generated from co-fed propylene were taken into account in the calculation of product distribution, leading to an overestimation of their selectivity. However, we could still observe that at constant propane conversion, decreasing WHSV resulted in increased selectivity of ethylene and CO and decreased selectivity for propylene. This indicated that lower WHSV benefited the deep-oxidation route. A brief discussion on this issue was added in page 10, line 157.

Supplementary Fig. 12 (a) C₃H₈ conversion as a function of temperature at various WHSV in “C₃H₈- C₃H₆” mode. (b) C₃H₆, (c) C₂H₄, and (d) CO selectivity as function of propane conversion at various WHSV in “C₃H₈- C₃H₆” mode. C₃H₈: C₃H₆: O₂: N₂ = 8: 2.5: 8: 21.5; reaction temperature: 430-500 °C.

11. Figure 2D is incredibly blurry and difficult to read.

a. What was happening with C₃H₆ during the pulse experiments? Given the hypothesis that a “co-reaction of the two hydrocarbons” is responsible for the observed C-C cleavage, C₃H₆ should also be consumed. We should see a similar consumption to what is shown in Supplementary Fig. 7 for ethylene.

b. Along the same lines, what does the production of C₃D₆ look like?

c. Was there any time delay in dosing and product formation? Especially given the hypothesis that PO(D) is a secondary product. There is no time scale on the x-axis, so I cannot tell for myself.

Reply: Figure 2D has been revised to improve its readability.

a. The signal of C₃H₆ was added in Figure 2D. As the reviewer said, the consumption of C₃H₆ and C₃D₈ increased with the increased temperature, indicating that the stronger synergy between C₃D₈ and C₃H₆ occurred at a higher temperature. This outcome was also consistent with the generated signal of C₂H₄. We added the corresponding discussion on page 11, line 179.

b. We have tried to track the C_3D_6 signal before, but no pattern was found, which may be caused by the overlapping signal of the m/z ratio between C_3D_6 and C_3D_8 .

c. Thank the reviewer for the valuable comment and we analyzed the pulse signal in more detail. As shown in Supplementary Fig. 14, the signal of C_3H_6 consumption and PO generation appeared simultaneously, indicating PO may be the primary product in the oxidation reaction of C_3H_6 . The PO(D) formation signal delayed the C_3H_6 consumption signal by 8.4 s, indicating PO(D) may be the secondary product that was generated from C_3D_8 -derived C_3D_6 . In addition, a similar phenomenon was also observed in the C_3D_8 pulse experiment co-fed with ethylene (Supplementary Fig. 16). A brief discussion was added in page 11, line 186, and page 12, line 193.

Supplementary Fig. 14 Gas profile in transient pulses of C_3D_8 with the environment for oxidation of propylene over BN at 490 °C. $F_{total} = 40 \text{ mL min}^{-1}$, $C_3H_6: O_2: N_2 = 2.5: 8: 29.5$; $m_{cat} = 100 \text{ mg}$; pulse value: 1 mL each time.

Supplementary Fig. 16 Gas profile in transient pulses of C_3D_8 with the environment for oxidation of ethylene over BN at 520 °C. $F_{total} = 40 \text{ mL min}^{-1}$, $C_2H_4: O_2: N_2 = 2.5: 8: 29.5$; $m_{cat} = 100 \text{ mg}$; pulse value: 1 mL each time.

12. Supplementary Fig. 7 is incredibly difficult to read.

Reply: Thank you for pointing it out. We have enlarged the font and thickened the line to improve the readability of **Supplementary Fig. 15** (Supplementary Fig. 7 in the previous version of the manuscript).

13. What are the reaction conditions used in Supplementary Fig. 8?

Reply: $F_{\text{total}} = 40 \text{ mL min}^{-1}$, $\text{O}_2: \text{N}_2 = 1: 4$; $m_{\text{cat}} = 100 \text{ mg}$; PO was bubbled into the reactor at $0 \text{ }^\circ\text{C}$. This information was added in the caption of **Supplementary Fig. 17** (Supplementary Fig. 8 in the previous version of the manuscript).

14. On pp. 13, lines 202-203, you state, “According to our experimental measurement, co-feeding of propylene results in a moderate increase of CO formation.” Where is this data shown? Please show the product distributions for the reactions you are comparing.

Reply: The product distribution was added in **Page 31** of supplementary information (**Supplementary Fig. 21**).

Supplementary Fig. 21 The products selectivity and propane conversion in “Single- C_3H_8 ” and “ C_3H_8 - C_3H_6 ” modes at $490 \text{ }^\circ\text{C}$. $F_{\text{total}} = 40 \text{ mL min}^{-1}$, “Single- C_3H_8 ”, $\text{C}_3\text{H}_8: \text{O}_2: \text{N}_2 = 8: 8: 24$; “ C_3H_8 - C_3H_6 ”, $\text{C}_3\text{H}_8: \text{C}_3\text{H}_6: \text{O}_2: \text{N}_2 = 8: 2.5: 8: 21.5$.

15. Pp. 14, line 220, “evolvment” should be “evolution”

Reply: Thank the reviewer for pointing it out. We have revised it in **page 17, line 275**.

16. Figure 4 is not at all clear:

- Which H is abstracted in the initial “H-abstraction” step? Given the different bond strengths of primary and secondary C-H bonds, I would expect different free energy barriers.
- Similarly, which C is the O bonding to? Does it make a difference?
- This proposed reaction network is not in line with the product distributions observed for propane ODH
- This proposed reaction network does not explain product distributions that have been previously reported in the literature. For example, according to Figure 4, C_2H_4 and CO are produced in a 1:1

ratio. Multiple studies report a higher ratio, indicating that there must be another route to ethylene formation that is not accounted for in the proposed network.

The proposed reaction network suggests that the only route to ethylene is the degradation of PO. If PO is pulsed into the reactor as done with other species in Fig. 2, do you observe an increase in ethylene?

e. In the degradation of $C_3H_6O \rightarrow C_2H_4 + CO$, what happens to the other 2 Hs?

Reply: a. We agree with the reviewer's comment. The energy barrier for initial H-abstraction corresponds to the primary C-H bond breaking, which is defined in the manuscript. The difference between the primary and secondary C-H bonds is small (Supplementary Fig. 22). Concerning the more presence of the primary C-H bond than that of the secondary C-H bond in propane (6:2), for simplicity, we focused on the activation of the primary C-H bond in the manuscript. A brief discussion was added in page 16, line 267.

Supplementary Fig. 22 The key stationary points and relative free energies (ΔG^\ddagger and ΔG , in kcal/mol) in the oxidation of propane by triplet O_2 to form peroxy radical in the gas phase.

b. According to our previous calculation work (*J. Phys. Chem. C*, 2021, 125, 24930–24944), the rate-determining step of alkoxy radical generation is the activation of the primary C-H (1i). The generated alkyl radical reacts with $\bullet OOH$ in a step-wise manner (two sequential steps with the formation of an intermediate (15ia) followed by the dissociation of the O-O peroxy bond (15ib)) or a concerted manner (15ii). The barrier to the concerted pathway is 20 kcal/mol (primary carbon), and 28.7 kcal/mol (secondary carbon), which is much lower than the activation energy barrier of

alkane (48.6kcal/mol). To avoid misunderstanding, we added a description of the barrier to the diagram in Figure 4 in the updated version: The barrier to the formation of the C-O bond is determined by the activation (H-abstraction) of propane. The barriers to the concerted reaction at the primary carbon and at the secondary carbon to generate alkoxy radical were added in Supplementary Fig. 23. A brief discussion was added in page 17, line 271.

Supplementary Fig. 23 The key stationary points and relative free energies (ΔG^\ddagger and ΔG , in kcal/mol) in the oxidation of propyl radical to form alkoxy radical in the gas phase.

c. We acknowledge and agree with the reviewer's comment. Due to the complexity and diversity of gas and surface reaction channels under ODH conditions, it is hard to directly relate the quantum chemistry calculation to the distribution of experimental reaction products. The present work focuses on the promotion effect of co-fed propylene, thus in the manuscript, we tried to limit our discussion of the reaction mechanism within the two key channels of propylene-promoted oxidative dehydrogenation of propane, which represent part of the ODH reaction network. We are now working on the mapping of the full reaction network, some of which were published in our previous work (*J. Phys. Chem. C*, 2021, 125, 24930–24944), and we will share our new findings when available.

d. The PO pulse experiment was carried out, and it was hard to collect the pulse signal even over a long period due to the much higher boiling point of PO compared with other gases. Thus QM calculations were carried out to complement our understanding of the role of PO in the reactions, based on which we proposed a mechanism in Figure 4 to elucidate the in situ formation of PO and

its participation in the ODH reaction. The elementary reaction steps in Figure 4 can explain the experimental results. We will extend our study on this issue from both the experimental side and computational sides and will share our new findings when available.

e. In the degradation of $C_3H_6O \rightarrow C_2H_4 + CO$, two H atoms of C_3H_6O may be abstracted by the oxidative radicals in its vicinity, e.g. OH and OOH, to assist its conversion to ethylene.

17. Figure 5 suffers from the same comparison issues that I noted earlier, as total [HC] differs pretty severely between the single-feed and co-feed experiments. Too many important variables change between experiments for the comparison to be meaningful.

a. Again, missing reaction details in figure caption.

Reply: According to the reviewer's suggestion, we conducted a “Single- C_2H_6 (10.5 sccm)” experiment where the HC: O_2 ratio was the same as that in the “ $C_2H_6-C_3H_8$ ” mode (Supplementary Fig. 29). In this scenario, the C_2H_6 conversion was lower than that in the “ $C_2H_6-C_3H_8$ ” mode and slightly higher than that in the “Single- C_2H_6 ” mode of the first version, indicating that the increased reactivity of C_2H_6 in “ $C_2H_6-C_3H_8$ ” mode was due to the addition of another alkane rather than changing the HC: O_2 ratio. We added the corresponding discussion in page 20, line 328. In the previous version of the manuscript, we kept specific HC contact times to guarantee a fair comparison of olefins production between the two feeding modes (mode 1: “ $C_2H_6-C_3H_8$ ”; mode 2: “Single- C_2H_6 ” + “Single- C_2H_8 ”) with the same amount of alkane inlet. Therefore, the previous comparison method in our work could also provide insight into the promotion effect of co-fed hydrocarbon.

Supplementary Fig. 29 C_2H_6 conversion in “Single- C_2H_6 (8 sccm)”, “Single- C_2H_6 (10.5 sccm)” and “ $C_2H_6-C_3H_8$ ” modes. $F_{total} = 40 \text{ mL min}^{-1}$. “Single- C_2H_6 (8 sccm)”, $C_2H_6: O_2: N_2 = 8: 8: 24$;

“Single-C₂H₆ (10.5 sccm)”, C₂H₆: O₂: N₂ = 10.5: 8: 21.5; “C₂H₆-C₃H₈”, C₂H₆: C₃H₈: O₂: N₂ = 8: 2.5: 8: 21.5.

a. Thanks for the reviewer’s reminder, we added reaction details in the figure caption.

18. Why are Figures 5E and 5F run at different temperatures?

Reply: We chose 520 °C as the reaction temperature for the cycling experiment of “Single-C₂H₆” or “C₂H₆-C₃H₈” to ensure that the carbon balance was above 95%.

19. Pp. 18 lines 278-280, “Remarkably, the added propane may not lead to irreversible positive or negative effects over BN catalyst because similar ethane conversions were obtained under three cycles of “Single-C₂H₆.” This observation is in line with Venegas, et al., who saw a similar promotional effect when co-feeding water in propane ODH over hBN (Angew. Chem. Int. Ed. 2020, 59, 16527.), which they similarly attributed to contributions in the gas phase, as water can enhance the radical pool concentration. It seems the added olefin is modifying the reaction similarly, although without the full product distributions, it is hard to say whether the addition of olefin enhances existing reaction pathways, or opens up additional reaction pathways. I’m curious to know what the product distributions obtained in the experiments here are.

Reply: As the reviewer said, similar promotional effects of water and olefins were observed in Venegas, et al. and our work. However, the effects of the two species were not exactly the same. First, for the work of Venegas, et al., the propane conversion steadily increased (during a wet cycle) or decreased (during a dry cycle) during the 12-hour cycling periods, indicating that the radical pool concentration was slowly increased and decreased, respectively. For the cycles of the “Single-C₂H₆” or “C₂H₆-C₃H₈” experiment, the C₂H₆ conversion rapidly increased and decreased within 25 min and remained steady, demonstrating that the concentration of active species enhanced by propylene could quickly reach a steady state. In addition, we added the data on product distribution in the “Single-C₃H₈” and “C₃H₈-C₃H₆” modes (Figure R2, R4). The ratio of CO_x to ethylene increased in the “C₃H₈-C₃H₆” mode, indirectly indicating the addition of propylene enhanced some deep-oxidation routes.

Figure R2 C₃H₈ conversion and product selectivity in “Single-C₃H₈” mode as a function of temperature over BN. $F_{\text{total}} = 40 \text{ mL min}^{-1}$, C₃H₈: O₂: N₂ = 8: 8: 24.

Figure R4 C₃H₈ and C₃H₆ conversion and product selectivity in “C₃H₈-C₃H₆” mode as a function of temperature over BN. $F_{\text{total}} = 40 \text{ mL min}^{-1}$, C₃H₈: C₃H₆: O₂: N₂ = 8: 2.5: 8: 21.5. The selectivity of C₃H₆ in the “C₃H₈-C₃H₆” mode was calculated from the net production of C₃H₆.

Overall, I think these results are very interesting and I look forward to seeing the improved manuscript! It's a truly fascinating system and your observation of PO formation is an important insight.

-Dr. Melissa Cendejas

Reply: We acknowledge the positive and constructive comments of the reviewer. We are very grateful to the reviewer for the high praise of our work. I hope we can work together to solve and explore the challenges and unknowns that await us in this fascinating boron-based catalytic system.

REVIEWER COMMENTS

Reviewer #1 (Remarks to the Author):

I have now had the chance to read the rebuttal letter, thereby addressing the comments of the three referees, as well as the full paper with the changes highlighted in yellow. Although I find the employed analytical methods still not that convincing and feel that some more work could have been done in this direction, I agree also with the other two referees that an interesting combination/system is made/proposed, which allows to produce light olefins from light alkanes over an attractive new catalyst material. Hence, based on this I recommend the work for publication in Nature Communications.

Reviewer #2 (Remarks to the Author):

I have carefully read the author's reply. Although a part of my concerns have been clarified I am unsure if the application potential of co-dehydrogenation of ethane and propane is advantageous from an industrial perspective and if the scientific impact of this study really reaches the level of Nature Communications. The remaining below concerns should be clarified.

(i) The authors have written "As shown in Supplementary Table 3, the selectivity for olefins in the co-feeding mode was essentially the same as that in the single-feeding mode. This shows the advantage of the co-feeding mode that high conversion can be achieved at lower reaction temperatures without significantly changing the selectivity of olefins." When the yield of ethylene and propylene in C₂H₆-C₃H₈, single C₂H₆ and single C₃H₈ is compared, the co-fed mode is disadvantageous. Moreover, a mixture of olefins is formed. Why is it advantageous to operate at lower temperatures in the case of an exothermic reaction? The challenge will be to remove the heat under industrially relevant conditions.

(ii) The presence of BO_x in differently activated samples simply means that a part of B in BN was oxidized. The role of BO_x was not elucidated to draw any conclusion about its relevance for the reaction studied.

(iii) When answering my question about propene over-oxidation, the authors refer to a paper published in Science Advances but ignore my original argument based on the first paper reported about the application of BN for the ODP reaction (Science (80-.). 354, 1570–1573 (2016).)

(iv) The authors have written "Owing to the complexity of the reaction system, which involves gas phase radical reactions and quenching, the conversion of propane does not linearly depend on catalysts loading" Why is the loading important when the same catalyst amount is used in a single-bed-reactor experiment and in a tandem-reactor experiment?

(v) The purpose of Figure B, C is unclear even after reading the author's reply.

(vi) The authors did not clarify my previous question about how propylene oxide can be "... formed by the interaction of propane and propylene in the ODH reaction of propane catalyzed by BN." Instead, they have written "PO may be the primary product in the oxidation reaction of C₃H₆." What is the role of propane in PO formation?

(vi) My original opinion about the mechanistic impact of Figure 4 was not changed.

Reviewer #3 (Remarks to the Author):

Thank you for your thorough responses to my initial review; the revised manuscript tells a stronger story. However, I have a couple lingering questions/comments before recommending publication:

1. In Figures 2d and S14, the C₂H₄ and PO signals both exhibit a small shoulder at an earlier reaction time that precedes the addition of C₃D₈ and consumption of C₃H₆. Can the authors comment on where this production might be coming from?

2. I agree that the effect of olefin addition is different from the effect of water addition into the reaction feed. However, I think we can also agree that it is fascinating that these two reaction products can have such a large effect on the conversion/reaction rate. I think the story you tell here would be more complete if there was some mention of the previously observed promotional effect of water on ODH over BN. I was surprised that in the introduction, H₂O was left off the list of reaction products (pg. 4, line 64)! As this work is building towards a more complete understanding of this complex reaction mechanism, it would benefit the author and reader if the promotional effect of water is mentioned. Perhaps such a discussion would inspire further studies of a potential synergy between the two promotional effects.

Response to the reviewers' comments

Reviewer #1

I have now had the chance to read the rebuttal letter, thereby addressing the comments of the three referees, as well as the full paper with the changes highlighted in yellow. Although I find the employed analytical methods still not that convincing and feel that some more work could have been done in this direction, I agree also with the other two referees that an interesting combination/system is made/proposed, which allows to produce light olefins from light alkanes over an attractive new catalyst material. Hence, based on this I recommend the work for publication in *Nature Communications*.

Reply: Thank reviewer for the positive recommendation. We will continue to make efforts in this direction.

Reviewer #2

I have carefully read the author's reply. Although a part of my concerns have been clarified I am unsure if the application potential of co-dehydrogenation of ethane and propane is advantageous from an industrial perspective and if the scientific impact of this study really reaches the level of *Nature Communications*. The remaining below concerns should be clarified.

Reply: We thank the reviewers for the comments. As a new ODH system discovered in recent years, many features of boron-based materials, such as nonmetallic active sites and surface-mediated gas-phase radical reaction mechanism, need to be constantly understood in term of fundamental research. Our work demonstrated an unusual phenomenon over boron nitride that in-situ formed olefins accelerated alkane conversion. We also proposed the possible reaction routes of the synergy between propane and propylene based on experimental evidence and DFT calculations. From the perspective of industrial application, we feel that two advantages in co-feeding alkane mixture can be considered: 1. The co-feeding mode can relieve the pressure of pre-separation of natural gas before reaction, which will reduce energy and carbon emissions; 2. The co-feeding mode can achieve the same level of alkane conversion and product yield at a lower temperature.

(i) The authors have written "As shown in Supplementary Table 3, the selectivity for olefins in the co-feeding mode was essentially the same as that in the single-feeding mode. This shows the

advantage of the co-feeding mode that high conversion can be achieved at lower reaction temperatures without significantly changing the selectivity of olefins.” When the yield of ethylene and propylene in C₂H₆-C₃H₈, single C₂H₆ and single C₃H₈ is compared, the co-fed mode is disadvantageous. Moreover, a mixture of olefins is formed. Why is it advantageous to operate at lower temperatures in the case of an exothermic reaction? The challenge will be to remove the heat under industrially relevant conditions.

Reply: The physical definition of yield is “conversion of the starting material to the desired product at fixed feed rate F , catalyst charge W , temperature T , pressure P , and feed composition C_0 ” (Leach, Bruce, ed. *Applied industrial catalysis*. Elsevier, 2012). In the “C₂H₆-C₃H₈” mode, ethylene could be formed from both ethane and propane, which would lead calculated yield to be overestimated or underestimated. For example, yield of C₂H₄ is 2.3% from C₂H₆ and 4.6% from C₃H₈; yield of C₃H₆ is 1.7% from C₂H₆ and 3.4% from C₃H₈. A reasonable comparison method would be to compare the space-time yield of olefins at the same alkanes WHSV (G. Ertl, *Handbook of Heterogeneous Catalysis*, Wiley-VCH, Weinheim, 2008). As shown in revised Supplementary Table 3, when the WHSV of ethane and propane was the same, space-time yield was similar in the co-feeding mode and single-feeding mode at the similar conversion. We added the corresponding equations in the supplementary information.

Supplementary Table 3 Alkane conversions, products distribution, and space-time yield at similar alkane conversions under different feeding modes.

Mode	weight hourly space velocity (WHSV, g g ⁻¹ h ⁻¹)			Conversion (%)			Selectivity (%)				Space-time yield (mmol g _{cat} ⁻¹ h ⁻¹)		
	C ₂ H ₆	C ₃ H ₈	Alkanes	C ₂ H ₆	C ₃ H ₈	Alkanes	C ₂ H ₄	C ₃ H ₆	CO	Olefins	C ₂ H ₄	C ₃ H ₆	Olefins
C ₂ H ₆ -C ₃ H ₈ (510 °C)	6.0	2.9	8.9	4.0	8.0	5.2	57.0	42.6	0.4	99.6	9.3	4.6	13.9
Single- C ₂ H ₆ (540 °C)	6.0	0	6.0	3.9	0	3.9	100	0	0	100	8.2	0	8.2
Single- C ₃ H ₈ (530 °C)	0	2.9	2.9	0	8.5	8.5	10.4	89.0	0.6	99.4	0.9	5.1	6.0
“Single- C ₂ H ₆ ” + “Single- C ₃ H ₈ ”	6.0	2.9	8.9	3.9	8.5	5.3	54.4	45.3	0.3	99.7	9.1	5.1	14.2

Indeed, in the co-feeding mode, a mixture of olefins is formed. In the heterogeneous catalysis system, the formation of mixture is a common phenomenon, thus product selectivity is an important

criterion to evaluate the efficacy of most reactions. Improving selectivity is the goal of researchers working in the field of catalysis, and more and more efficient catalyst structures and processes are being designed and developed to move toward this goal.

In an exothermic reaction with thermodynamic equilibrium, based on Le Chatelier's principle, the advantage of operating at a lower temperature is that an equilibrium can be expected to shift in the product direction (P Atkins, JD Paula, PW Atkins. *Physical chemistry: Thermodynamics, structure, and change*, 2014). The ODH reaction is an exothermic reaction that is not constrained by thermodynamic equilibrium. There are three other advantages for operating at lower temperatures in the ODH reaction: 1. Suppressing hot spot formation caused by the higher temperature; 2. Reducing energy consumption; 3. Reducing the risk of explosion in the atmosphere containing oxygen.

We agree with the reviewer's opinion that removing the heat under industrially relevant conditions is important. Compared with state-of-the-art metal oxide catalysts, BN can maintain a more uniform temperature profile because of its high thermal conductivity (33 W/m K), which awards it with remarkable ability for heat removal (*Chem. Eng. Sci.*, 2018, 186, 142-151).

(ii) The presence of BO_x in differently activated samples simply means that a part of B in BN was oxidized. The role of BO_x was not elucidated to draw any conclusion about its relevance for the reaction studied.

Reply: Thank the reviewer for kind comment. It has been widely accepted that the BO_x species in BN are active sites in the ODH reaction (*J. Am. Chem. Soc.*, 2019, 141, 182–190; *Chem. Soc. Rev.*, 2021, 50, 1438–1468; *ACS Catal.*, 2021, 11, 9370-9376). In addition, an important role of BO_x species is considered to generate the highly active radicals that take part in the chain dehydrogenation reaction in the gas phase (*Angew. Chem. Int. Ed.*, 2020, 59, 16527-16535; *J. Phys. Chem. C*, 2021, 125, 24930-24944). Thus, our analyses were based that BO_x species were able to produce these radicals. We added some discussions in page 13, line 222 for elucidating the role of BO_x species.

In our work, the content of BO_x species were similar in different activated samples, excluding the possibility that the different activity in single-feeding and co-feeding modes was due to the number of active sites. We added this discussion in page 8, line 125.

(iii) When answering my question about propene over-oxidation, the authors refer to a paper published in Science Advances but ignore my original argument based on the first paper reported about the application of BN for the ODP reaction (Science (80-.). 354, 1570–1573 (2016).)

Reply: Thank the reviewer for constructive suggestion. Indeed, the *Science* paper (2016) reported “Use of BN materials results in extraordinary selectivity to propene, among the highest reported under ODHP conditions. For instance, h-BN afforded 79% selectivity to propene at 14% propane conversion (Figure R1). Meanwhile, the traditional V/SiO₂ allows for a modest 61% propene selectivity at only 9% propane conversion (12). The selectivities obtained by using state-of-the-art ODHP catalysts (11, 13–19) are compared in Fig. 1A. The decrease in propene selectivity with increasing propane conversion is indicative of the facile overoxidation of propene to CO_x. Comparisons between key reaction parameters of the referenced catalysts are included in table S1”. As we have understood, the propylene selectivity over BN is much better than that of other catalysts, but slightly decreasing as the conversion rate increases, implying that perhaps over-oxidation routes is occurred. In our work, we confirmed the existence of over-oxidation routes through direct isotope experimental evidence. We modified the statements in page 1, line 16 and page 5, line 75 to avoid misunderstanding and added more discussions about the above *Science* paper in page 12, line 192. Thank the reviewer again.

Figure R1 Selectivity to propene plotted against propane conversion for ODHP, comparing previously reported data from representative catalysts to hexagonal boron nitride (h-BN) and boron nitride nanotubes (BNNTs). (*Science*, 2016, 354, 1570-1573)

(iv) The authors have written “Owing to the complexity of the reaction system, which involves gas phase radical reactions and quenching, the conversion of propane does not linearly depend on

catalysts loading” Why is the loading important when the same catalyst amount is used in a single-bed-reactor experiment and in a tandem-reactor experiment?

Reply: Thank the reviewer for constructive comment. The different activity in various loading modes is caused by the special reaction mechanism of BN which combines surface and gas phase reactions. The bed volume and post-catalytic volume were different in single-bed reactor and tandem reactor, resulting in different performances. The effects of bed volume and post-catalytic volume on catalytic activity have also been reported in previous studies (*Org. Process Res. Dev.*, 2018, 22, 1644-1652; M. Nadjafi, research-collection.ethz.ch). We added some discussions in page 6, line 97.

(v) The purpose of Figure B, C is unclear even after reading the author’s reply.

Reply: We are sorry for making “the purpose of Figure B, C” unclear still in the first reply. The purpose of Figure B, and C is to prove that the synergistic effect between alkane and olefin is more obvious in a longer contact time. After minutely consider reviewer comment, we reply as below and added the discussion in page 10, line 157, hopefully reviewer would satisfy with our effort. “*As the WHSV decreased, a decrease in the selectivity of ethylene and an increase in the selectivity of CO at the same propane conversion was observed (Figure 2b, c). It may be ascribed to that more part of the propane conversion was contributed by the interaction between ethylene and propane at a lower WHSV, leading to more ethylene conversion to CO. However, the propylene selectivity was insensitive to the change in WHSV (Supplementary Fig. 11b), which may be because propane could produce additional propylene to compensate for the reduced selectivity when interacting with olefins, resulting in almost constant selectivity of propylene.*” Based on the above outcome, we could indirectly deduce that the synergistic effect may occur in the propane ODH reaction, and this effect was further confirmed in the subsequent isotope experiments.”

(vi) The authors did not clarify my previous question about how propylene oxide can be “... formed by the interaction of propane and propylene in the ODH reaction of propane catalyzed by BN.” Instead, they have written “PO may be the primary product in the oxidation reaction of C₃H₆.” What is the role of propane in PO formation?

Reply: We are sorry that our first reply is a bit simple about the role of propane in PO formation. According to our DFT calculation, PO can form via two paths in the presence of propene: C₃H₆ + OOH → PO + OH; C₃H₆ + OC₃H₇ → PO + C₃H₇. In the second pathway, the propoxyl radical was

obtained by oxidizing propane. This indicates that the propane was oxidized first to propyl radical, which then converted to propoxyl radical in the presence of dioxygen. These two paths are energetically feasible under the ODH conditions, demonstrating the cooperation of propene and propane in the formation of PO. Our pulse experiment showed that PO was formed from propylene when adding propane, also supporting the DFT result that propane could enhance the conversion of propylene to PO. There could be other pathways to form PO. The study of mechanism in this system is still an open question, which is worth further discussing. We have added more discussion in page 17, line 277.

(vi) My original opinion about the mechanistic impact of Figure 4 was not changed.

Reply: The reviewer’s original opinion was “Figure 4 is incomplete and speculative”. We agree with the reviewer, and minutely modify the Figure 4 in revised version to avoid possible confusion. The corresponding caption have been modified in page 19, line 299. And we have changed “reaction network” to “reaction routes”. In the figure, we showed the formation of $\bullet\text{OOH}$ and $\bullet\text{OC}_3\text{H}_7$, and their reactions with C_3H_6 to produce PO, which is the message we want to share with the community. As we can see, the reactions of propylene with peroxy and propoxyl radical species produce PO, and in the meantime, release hydroxyl and propyl radicals. This figure demonstrates the promotion role of propene in the conversion of propane. We appreciate reviewer very much for all serious-minded comments.

Figure 4. Reaction routes associated with the gaseous interactions between propane and propylene in the ODH reaction.

Reviewer #3

Thank you for your thorough responses to my initial review; the revised manuscript tells a stronger story. However, I have a couple lingering questions/comments before recommending publication:

1. In Figures 2d and S14, the C_2H_4 and PO signals both exhibit a small shoulder at an earlier reaction time that precedes the addition of C_3D_8 and consumption of C_3H_6 . Can the authors comment on where this production might be coming from?

Reply: As the reviewer mentioned, we also observed the small shoulder peak during the isotope experiments (Fig 2d and S14). Through amplifying the signal and comparing the intensity of peak (Figure R2), we found: 1. Decrease of C_3D_8 signal was observed before the main signal, and the descent started and end at the same time as the shoulder. 2. The height of the shoulder was the same as the baseline after the pulse. According to the above two phenomena, we deduced this small shoulder is a signal fluctuation by inserting the syringe into the channel prior to pulse. We added the corresponding discussion in page 12, line 196.

Figure R2. Signals of C_3D_8 , C_2H_4 , and PO when pulsing C_3D_8 at 490 °C.

2. I agree that the effect of olefin addition is different from the effect of water addition into the reaction feed. However, I think we can also agree that it is fascinating that these two reaction products can have such a large effect on the conversion/reaction rate. I think the story you tell here would be more complete if there was some mention of the previously observed promotional effect of water on ODH over BN. I was surprised that in the introduction, H_2O was left off the list of reaction products (pg. 4, line 64)! As this work is building towards a more complete understanding

of this complex reaction mechanism, it would benefit the author and reader if the promotional effect of water is mentioned. Perhaps such a discussion would inspire further studies of a potential synergy between the two promotional effects.

Reply: Thanks for constructive suggestion. Indeed, water could promote the propane conversion in BN-catalyzed ODH reaction. To exclude the interference, in our experiments, H₂O was removed before the exhaust of R1 entered the R2. Therefore, H₂O was left off the list of reaction products (pg. 4, line 64). For clarity, we added a cold trap in the diagram of Figure 1 and added an explanation in page 4, line 67 in revised version. As suggested, in order to benefit the author and reader, the promotional effect of water as well as the different role in olefin and water were mentioned in page 25, line 405 in revised version. *“In addition to olefins, H₂O has been shown to improve the BN-catalyzed ODH performance, which is another product in the reaction.¹⁸ Because two stages of rapid and slow changes of alkane conversion were observed in the cyclic experiment, two possible roles of H₂O were proposed to generate free radicals and increase the concentration of active surface species. Our previous work on calculation also supported that H₂O could interact with the surface to form more B-OH.¹⁹ Compared with H₂O, propylene only showed a rapid change of alkane conversion in the cyclic experiment. In addition, the presence of olefins may enhance the deep-oxidation route while H₂O did not. Therefore, although both species demonstrated the promotional effect on alkane conversion, they differed in enhancing reaction routes and changing the species or concentration of free radicals and surface active sites. These observations suggest that the complex surface-gas phase reaction involved many synergies and it is worth further investigation in the future to comprehend this system.”*

REVIEWERS' COMMENTS

Reviewer #2 (Remarks to the Author):

The author's reply to my remaining comments and the changes made in the manuscript and the SI are constructive and clarify some of the concerns. I am, however, still unsure if the scientific impact of this study is really significant. As seen in Supplementary Table 3, there are no improvements in the selectivity to C₂H₄+C₃H₆ and in the space time yield of their formation when comparing the results obtained using a feed containing C₂H₆ and C₃H₈ with those using single C₂H₆ plus single C₃H₈ feeds. Under this consideration it is unclear why this study "... provided an insight into efficiently producing olefins" as written in the abstract. The possibility to operate at 20 or 30°C lower temperatures in comparison with individual oxidative dehydrogenation of propane or ethane is not a big advantage. In summary, despite my remaining concerns, I would agree with other reviewers when they recommended the manuscript for publication.

Reviewer #3 (Remarks to the Author):

I have read through the revised manuscript and responses to reviewers. I thank the authors for their efforts in revising this work; I believe this current version makes a much stronger and more compelling story than the original submission. The results presented here add crucial information to the quickly evolving understanding of boron-based catalysis. I recommend this work for publication in Nature Communications.